# The Importance of Plume Rise on the Concentrations and Atmospheric Impacts of Biomass Burning Aerosol

Carolin Walter[1], Saulo R. Freitas[2,3], Christoph Kottmeier[1], Isabel Kraut[1], Daniel Rieger[1], Heike Vogel[1], and Bernhard Vogel[1]

[1]Karlsruhe Institute of Technology, Institute for Meteorology and Climate Research, Karlsruhe, Germany
[2]CPTEC Center for Weather Forecasts and Climate Studies, National Institute for Space Research, Cachoeira Paulista, Brazil
[3]now at: NASA Goddard Space Flight Center & USRA/GESTAR, Greenbelt, Maryland, USA

*Correspondence to:* Carolin Walter (carolin.walter@kit.edu)

**Abstract.** We quantified the effects of the plume rise of biomass burning aerosol and gases for the forest fires that occurred in Saskatchewan, Canada, in July 2010. For this purpose, simulations with different assumptions regarding the plume rise and the vertical distribution of the emissions were conducted. Based on comparisons with observations, applying a one-dimensional plume rise model to predict the injection layer in combination with a parametrization of the vertical distribution of the emissions

outperforms approaches in which the plume heights are initially predefined. Approximately 30 % of the fires exceed the height of 2 km with a maximum height of 8.6 km. Using this plume rise model, comparisons with satellite images in the visible spectral range show a very good agreement between the simulated and observed spatial distributions of the biomass burning plume. The simulated AOD with data of an AERONET station is in good agreement with respect to the absolute values and the timing of the maximum. Comparison of the vertical distribution of the biomass burning aerosol with CALIPSO retrievals also

showed the best agreement when the plume rise model was applied. We found that downwelling surface short-wave radiation below the forest fire plume is reduced by up to 50 % and that the 2 m temperature is decreased by up to 6 K. In addition, we simulated a strong change in atmospheric stability within the biomass burning plume.

## 1 Introduction

Emissions from biomass burning significantly contribute to the global atmospheric aerosol mass (Stocker et al., 2013). A total

aerosol particle mass of 1.33 Tg is produced by vegetation fires each year, which implies a release of 12.7 Gg of particles with diameters of less than 2.5 $\mu$m (Zhang et al., 2012). Biomass burning aerosol degrades air quality (Ignotti et al., 2010), impacts aviation by reducing visibility, modifies the radiative fluxes and has an impact on cloud microphysics and precipitation. Therefore, there is an urgent need to forecast the dispersion of biomass burning aerosol and its impact on the state of the atmosphere on the time scale of days. This is currently not performed in the majority of operational forecast centres (i.e.

weather services), with few exceptions, such as the Brazilian center CPTEC/INPE, which has been running operationally since 2003 (Freitas et al., 2005). However, quantifying the effects of biomass burning aerosol on cloud formation and precipitation is still an unresolved scientific problem.

In the summer of 2002, the U.S. east coast experienced some of the worst air quality days during the passage of biomass burning plumes (Colarco et al., 2004). The emissions were released into the air over central Quebec, Canada, and further transported with a descending air mass prior to being mixed into the boundary layer of Washington D.C., USA. Another example of extreme air pollution are the Russian wildfires in 2010, which caused a very high aerosol concentration in the Moscow region and dramatically reduced visibility (Konovalov et al., 2011). During the summer of 2010, the Russian fires emitted approximately 10 Tg of CO, which is more than 85 % of the annual anthropogenic CO emissions in this region. In their study, Konovalov et al. (2011) developed an optimization procedure to quantify the wildfire emissions. As a result, their simulated concentrations of CO and PM10 were well correlated with the measured concentrations.

The radiative impact of biomass burning aerosol depends on the size distribution and optical properties of the particles. Fires emit gaseous precursors of aerosol and primary particles such as soot. The optical properties of soot produced by different fuels were investigated by Colbeck et al. (1997). Flaming combustion at high temperatures was found to produce smoke, which strongly absorbs solar radiation. Smouldering combustion at lower temperatures produces aerosol that predominantly scatters light. Incomplete combustion leads to smoke particles that contain a substantial fraction of unburned organic matter. The presence of more elemental carbon leads to blacker smoke. Furthermore, the ageing of the particles leads to a change in chemical composition and a shift in the size distribution due to coagulation processes, which influence the radiative properties (Sakamoto et al., 2015). Local heating due to the absorption of radiation results in the evaporation of cloud droplets and in the dissolution of clouds. This is called the semi-direct aerosol effect (Ackerman et al., 2000; Lohmann and Feichter, 2001). Depending on their size distribution and chemical composition, the aerosol particles can also act as cloud condensation nuclei or as ice nuclei. Consequently, aerosol particles might change cloud properties such as albedo, lifetime and precipitation efficiency.

Koren et al. (2004) utilised MODIS data to analyse cloud cover under smoky conditions. The stable meteorological conditions and homogeneous cloud distribution in the Amazon without the presence of smoke provided an ideal basis for investigating the impact of biomass burning aerosol on cloud formation. They determined a mean cloud cover reduction of 50 % for an Aerosol Optical Depth (AOD) of 0.6. An increase in the AOD led to a further reduction in cloud cover. When the AOD increased above 1.3, no more clouds were present. Biomass burning aerosol is also known to have an influence on precipitation and to cause hail events, which are not reported for smoke-free conditions (Ding et al., 2013; Andreae et al., 2004).

Ding et al. (2013) explicitly investigated the influence of extreme air pollution due to biomass burning on weather forecasting. A temperature of 34 °C was predicted for Nanjing, China, for 10 June 2012, but the temperature only reached 26.5 °C. The high concentration of scattering and absorbing aerosol reduced the solar radiation by 70 %. This influence of high pollution conditions on weather is proofed by the fact that under unpolluted conditions on the previous days, the model forecast agreed well with observations. During such events, the forecast quality of numerical models was dramatically decreased because scattering and absorbing biomass burning particles were not considered.

In the case of forest fires, not only the total emitted mass is of importance but high amounts of energy are also released into the atmosphere. Thus, the buoyancy of the plume becomes important for the injection height. Due to the heat released by the fires and the temperature difference between the plume and the environment, an updraft is initiated that transports the

emitted gases and aerosols to higher altitudes. The height at which the emissions are effectively released is critical for a reliable simulation of the transport of the emitted chemical species. In the free atmosphere, aerosol has a considerably longer lifetime. Moreover, the aerosol is quickly transported out of the source region by the prevailing wind. The effective source height defines the height below which the emitted material is released in a model system to represent the real plume. The effective source height depends on the heat released by the fire and on the environmental conditions: temperature, stability, humidity and wind speed (Penner et al., 1986). Additional buoyancy can be gained through the release of latent heat by condensation, whereas a strong horizontal wind prevents the air parcel from reaching the condensation level (Freitas et al., 2007). The release of latent heat from large fires can make an important contribution to the formation of pyrocumulus and pyrocumulonimbus clouds (Fromm et al., 2010).

Observations show that smoke plumes of biomass burnings commonly reach the free troposphere or even exceed it and penetrate the stratosphere (Andreae et al., 2004; Trentmann et al., 2006). In some cases, pollutants originating from Canadian forest fires have even been detected over Europe. This is due to lifting into high altitudes by pyro-convection prior to horizontal advection over the North Atlantic Ocean (Fiebig et al., 2003; Waibel et al., 1999). In the standard version of GEOS-Chem (Goddard Earth Observing System with Chemistry), pyrogenic emissions are released at the model surface (Mu et al., 2011). Assuming that the emissions are dispersed within the boundary layer is only sufficient for small fires (Tosca et al., 2011). A common method in aerosol transport modelling is to assume a vertically homogeneous distribution of the fire emissions from the ground to a certain height (Wang et al., 2006). For example, Pfister et al. (2005) utilised a vertically homogeneous distribution between the surface and 400 hPa for simulations in Alaska. Wang et al. (2013) simulated the transport of smoke from fires over the South-east Asian Maritime Continent for September and October 2006. They concluded that in comparison with CALIOP (Cloud-Aerosol Lidar with Orthogonal Polarisation) observations simulations with homogeneous emissions between the surface and 800 m provided the best agreement for the aerosol profile in their case. The simulation with homogeneous emissions between the surface and 2 km showed a rather different picture than the CALIOP profile. The same effective source height of 800 m was used by Ge et al. (2014). However, observations with the Multi-angle Imaging SpectroRadiometer (MISR) on-board the NASA Terra satellite showed that with sufficient buoyancy, the aerosol is not distributed homogeneously up to a certain height (Kahn et al., 2007; Val Martin et al., 2010). Moreover, the aerosol is concentrated in a discrete, elevated layer of relative stability after being transported through the boundary layer by the initial buoyancy. According to observations by Kahn et al. (2007), this layer is located between 5 and 6.5 km. Aerosol also concentrates at 1 km when originating from sources with less initial buoyancy. Smoke layers in a stable stratified atmosphere are not as deep as layers in an unstable stratified atmosphere (Val Martin et al., 2010). Val Martin et al. (2010) indicated that both the fire intensity and the stability of the atmosphere are crucial for the effective source height. A recent review of the representation of plume injection heights in atmospheric models was performed by Paugam et al. (2016).

Including a one-dimensional, subgrid-scale plume rise model into an existing chemistry and transportation model has the potential to enhance the representation of fire emissions in model simulations (Freitas et al., 2007; Sessions et al., 2011). The one-dimensional model calculates the upper and lower bounds of the injection layer depending on meteorological conditions, fire size and fire intensity for every fire location. The emitted material is released between these bounds (Freitas et al., 2007).

Sessions et al. (2011) simulated the source height with the one-dimensional plume rise model implemented in WRF-Chem (Weather Research and Forecast modeling system coupled with chemistry). The source heights of the plume rise model are in good agreement with satellite-retrieved source heights. The model results are compared with basic distributions such as injection in the boundary layer and injection in the layer between 3-5 km. The conclusion is that different source heights lead to different transportation paths.

Sofiev et al. (2012) used Convective Available Potential Energy (CAPE) calculations to constrain plume heights. In this method dynamical entrainment is neglected. The plume height depends on stratification, turbulent entrainment, fire size and heat release. The function for the plume height was fitted to MISR fire observations.

Another possibility to account for the fire-induced buoyancy in the determination of the effective source height is to decrease the grid spacing, thereby allowing the buoyancy-driven dynamics and the resulting convective plumes to be explicitly solved by the model. Trentmann et al. (2002) simulated a fire experiment in Quinault (Washington, USA) in 1994 using the Active Tracer High-resolution Model (ATHAM). The model domain had a size of 35x28 $km^2$ and a vertical extension of 3.75 km, with a minimum horizontal grid spacing of 50 m and in vertical 20 m at the centre. This model was also applied for simulating the pyroconvection of the Chisholm fire in Alberta, Canada, in 2001 (Luderer et al., 2006; Trentmann et al., 2006). This fire was a strong event with emissions that were transported up to the stratosphere.

In this study, we extend the comprehensive online-coupled model system COSMO-ART (Vogel et al., 2009) to quantify the influence of biomass burning aerosol on the state of the atmosphere. To account for the fire-induced buoyancy of fire emissions, we incorporate the one-dimensional plume rise model of Freitas et al. (2006). We use the daily GFASv1.1 (Global Fire Assimilation System, Kaiser et al., 2012) datasets for the fire emissions. A parametrization of the diurnal cycle for fires in boreal forests is developed. With this framework, we are able to consider chemical processes such as particle formation and the photochemical ageing of the released particles. As a case study, we simulate a fire event that occurred in July 2010 in the north of Saskatchewan, Canada. Several simulations with different setups are conducted to evaluate the importance of plume height on the smoke distribution. Furthermore, we investigate the radiative impact of the biomass burning aerosol. To the best of our knowledge we are the first to investigate the effect of biomass burning aerosol on temperature and dynamics with an online-coupled modelling system on synoptic time scales with an explicit treatment of the aging of soot in combination with a plume rise model.

In section 2, a short description of COSMO-ART is presented. Then, the one-dimensional plume rise model is explained. The results of a sensitivity study are presented to motivate the coupling of a plume rise model with a comprehensive model system. The emission database is described, and the optical properties of the emissions are discussed. In section 3, the simulation results are presented and evaluated. A comparison with observations is given, and the impacts of the biomass burning emissions on radiation and temperature are presented.

## 2 Model description

The simulations are conducted using the comprehensive online-coupled model system COSMO-ART (Consortium for Small-scale Modelling - Aerosols and Reactive Trace gases, Vogel et al., 2009). This system is based on the operational weather forecast model COSMO (Baldauf et al., 2011). COSMO-ART includes a comprehensive chemistry module to describe the gaseous composition of the atmosphere and secondary aerosol formation, and it allows for feedback of the simulated aerosol particles with radiation, cloud formation, and precipitation (Stanelle et al., 2010; Knote et al., 2011; Bangert et al., 2012; Lundgren et al., 2013; Athanasopoulou et al., 2014; Rieger et al., 2014; Vogel et al., 2014).

The size distribution of aerosol within COSMO-ART is approximated by log-normal distributions. In Table 1, all required modes with their initial median diameters, standard deviations and chemical compositions are presented. The standard deviation is maintained constant while the median diameter of the aerosol changes during transport. Chemical reactions are calculated with RADMKA (Regional Acid Deposition Model Version Karlsruhe, Vogel et al., 2009) which is based on RADM2 (Regional Acid Deposition Model, Stockwell et al., 1990). The formation of secondary organic aerosol in calculated by a VBS approach (volatility basis set, Athanasopoulou et al., 2012). COSMO-ART explicitly treats the aging of soot particles transferring them from external to internal mixtures as described in Riemer et al. (2003). The radiative fluxes are calculated with the GRAALS radiation scheme (Ritter and Geleyn, 1992). A priori Mie-calculations have been performed for the initial aerosol particle size distributions and their chemical composition to obtain mass specific values for the extinction coefficient, single scattering albedo, and asymmetry parameter. These coefficients also depend on wavelength. To consider the optical properties of the current aerosol distribution the mass specific parameters obtained by the Mie-calculation are weighted with the mass fraction of the chemical components. Within COSMO-ART a full two-moment cloud microphysics scheme (Seifert and Beheng, 2006) is used. Aerosol activation is considered according to Fountoukis and Nenes (2005). Ice nucleation is based on the parameterization by Phillips et al. (2008). Cirrus formation and the competition between homogeneous and heterogeneous freezing is specified according to Barahona and Nenes (2009a, b).

### 2.1 Plume rise model

The model system COSMO-ART is extended by the one-dimensional, sub-grid-scale plume rise model of Freitas et al. (2006, 2007, 2010), which is briefly summarised in the following. The plume rise model accounts for buoyancy, atmospheric stratification, and flow conditions to calculate the plume height. The relevant processes occur on a considerably smaller scale than the grid spacing of regional modelling systems such as COSMO-ART. The one-dimensional plume rise model uses an internal vertical grid spacing of 100 m with 200 vertical layers. The environmental conditions, such as pressure, humidity, temperature and wind speed, are calculated by COSMO-ART. Hourly, for every grid point with an active fire, the values of these variables are transferred to the plume rise model in order to calculate the current plume height. Within an hour, the plume height is maintained constant. The parameters fire size and fire intensity depending on vegetation type and density have to be set. With these variables, the heat release and the initial buoyancy are calculated. The lower boundary condition is based on the assumption of a virtual source of buoyancy below the surface, resulting in a high vertical velocity at the surface. The final buoyancy

is limited by turbulent and dynamic entrainment. Turbulent entrainment leads to dilution and an increased plume radius. The latter accounts for the environmental wind speed and the possibility of a bent-over of the smoke plume. The buoyancy is increased by latent heat release when condensation occurs. By definition, the top of the plume is reached as soon as the vertical velocity decreases below 1 m s$^{-1}$. To achieve upper and lower bounds of the effective emission layer, two values are chosen for the parameter fire intensity. This is a robust approach for obtaining a vertical extended layer with the advantage that the environmental conditions control the extension. For the upper bound, we specified 80 kWm$^{-2}$. For the lower bound, the value is 30 kWm$^{-2}$. The fire area is set to 50 ha. In this way, we obtain a buoyancy-dependent thickness of our layer. The application of a range of heat flux is justified not only by the variability associated with the vegetation condition, which is not known, but also by the own dynamic variation during the combustion process. Besides, this range is also applied in a statistical sense since the net emission in the three-dimensional atmospheric transport model might be associated not with a unique fire but a set of sub-grid scale fires all burning inside the same model grid box. Using the Fire Radiative Power (FRP) to estimate the buoyancy flux does not help to eliminate the use of the prescribed range of the heat flux, since there is still a substantial uncertainty in converting FRP to the convective energy, which has been widely described in the literature (Wooster et al., 2005; Val Martin et al., 2012; Paugam et al., 2015). Moreover, the uncertainty in the FRP retrieval by sensors on-board of satellites is also high.

To demonstrate the importance of meteorological conditions on the maximum height of the plume top, a sensitivity study was performed. This was done for a fixed fire size and a fire intensity of 80 kWm$^{-2}$ for an imaginary fire in central Saskatchewan. The results are summarized in Fig. 1. We prescribed the profile of the potential temperature (Fig. 1a) and the horizontal wind speed (Fig. 1b) for 16 and 17 July 2010 at 18:00 UTC as the environmental conditions. The buoyancy calculated by the plume rise model for these conditions is depicted in Fig. 1c, and the resulting vertical velocity is shown in Fig. 1d. The top of the plume on 16 July 2010 is reached at an altitude of 5300 m. The next day, the plume does not exceed 2200 m. The horizontal wind speed on 17 July is at least twice as high as that on 16 July, leading to a strong dynamical entrainment that decreases the buoyancy. The potential temperature indicates a stable stratification for 16 July. On 17 July, the stratification is strongly unstable at the surface and is becoming more neutral up to 1200 m. Thus, the initial buoyancy is considerably greater in this case. The buoyancy decreases very strongly with increasing horizontal wind speed, but the plume arises out of the unstable layer because of its inertia. Although the buoyancy on 16 July has a smaller initial value and reaches its minimum at lower altitude, the resulting maximum plume height is greater. This result is due to the additional buoyancy gained by latent heat release from condensation starting at an altitude of approximately 1200 m on this day. This single example already shows the substantial impact of meteorological conditions on plume rise.

## 2.2 Vertical distribution of the emissions

The emissions are distributed with a parabolic function defined between the upper and the lower bounds as specified within the plume rise model and according to the following expression:

$$f(z^*) = 6 \cdot z^* \cdot (1 - z^*). \tag{1}$$

The dimensionless height $z^*$ is determined by

$$z^* = \frac{z - z_{bot}}{z_{top} - z_{bot}}, \tag{2}$$

where $z$ is the model layer height and $z_{top}$ and $z_{bot}$ are the plume tops determined by the plume rise model using the upper and lower limits for the heat flux, describing the upper and lower bounds of the emission layer. Consequently, we obtain

$0 \leq z^* \leq 1$. Implying a given total emission $Q_{tot}$, the emission profile can be calculated depending on $z^*$:

$$Q(z^*) = \begin{cases} Q_{tot} \cdot f(z^*) & \text{if } 0 < z^* < 1 \\ 0 & \text{else.} \end{cases} \tag{3}$$

To obtain the portion of the total emissions that is released in the respective grid cell, the parabolic function is integrated from the lower level of the grid cell ($z^*_{lolev}$) to the upper level of the grid cell ($z^*_{uplev}$):

$$\int_{z^*_{lolev}}^{z^*_{uplev}} f(z^*) dz^* = \left[ 3z^{*2} - 2z^{*3} \right]_{z^*_{lolev}}^{z^*_{uplev}}. \tag{4}$$

## 2.3   Diurnal cycle

The intensity of vegetation fires depends on the time of day. This dependency is subject to the diurnal cycle of meteorological variables such as temperature, humidity and wind speed. For ignition, the humidity of the biomass load has to be 35 % or less. During daytime, the humidity at the surface is reduced due to solar irradiation and increasing turbulent mixture, therefore, ignition and fire spread are favoured. The spread is also advanced due to a higher wind speed (McRae et al., 2005). The

wind speed in the boundary layer is usually higher during daytime than at night. Accordingly, fires have a diurnal cycle with maximum emissions in the early afternoon and minimum emissions at night (Prins et al., 1998). Wang et al. (2006) claimed that the application of hourly emission data is crucial for a realistic dispersion of the smoke near the source region. With geostationary satellites, it is possible to characterize the diurnal cycle of the fires (Kaiser et al., 2009b; Andela et al., 2015). Zhang and Kondragunta (2008) analysed the daily variability by considering variations of the fire pixel size. The diurnal cycle

is not only a function of meteorological conditions but also dependent on the vegetation type of the burning biomass (Giglio, 2007). According to Zhang and Kondragunta (2008), the daily maximum of the fire pixel size is reached between 10:00 and 15:00 local solar time (LST). During this period, 52.1 % of the daily amount of emissions are released in a forest. In the off-peak time, the hourly emissions are 2-4 % of the daily amount. A diurnal cycle may be specified through a weighted normal distribution (Kaiser et al., 2009a).

$$d(t_l) = w + (1 - w) \frac{24h}{\sigma \sqrt{2\pi}} exp \left( -\frac{1}{2} \left( \frac{t_l - t_0}{\sigma} \right)^2 \right) \tag{5}$$

where w is the weighting, $t_l$ denotes the local solar time, $t_0$ is the expected value, and $\sigma$ is the standard deviation. We designate the values as $w = 0.3$, $t_0 = 12.5\,h$ and $\sigma = 2.5\,h$. Subsequently, the equation expresses the diurnal cycle for forest fires as described by Zhang and Kondragunta (2008), and it is shown in Fig. 2. This diurnal cycle is overlaid on the mean values of fire size and fire intensity used in the plume rise calculation. Within an hour the emissions are maintained constant.

## 2.4 Radiative properties

Calculating the radiative effect of biomass burning and other aerosol types requires the optical properties extinction coefficient, single scattering albedo, and asymmetry parameter of the aerosol particles at each grid point and each time step. These optical properties depend on the refractive index of the individual compounds, the chemical composition of the particles, their shape, and their size distribution. The refractive index and therefore the optical properties depend on the wavelength. Insoluble light absorbing particles like soot can be covered by a soluble shell due to physical (coagulation, condensation), and photo-chemical ageing. This increases their mass absorption efficiency (Riemer et al., 2003; Saleh et al., 2014; Bond et al., 2013). That effect needs to be accounted for in fully online-coupled model systems like COSMO-ART. Mie-calculations are the adequate method to determine the optical properties from given size distributions and their chemical composition (Bohren and Huffman, 1983). These calculations are very time consuming and therefore it is not possible to perform them at each grid point and at each time step. Instead, we have developed a parameterization as described in Vogel et al. (2009). This parameterization is based on simulated aerosol distributions and detailed Mie-calculations ending in mass specific values of the extinction coefficient, single scattering albedo and asymmetry parameter. Moreover, this parameterization takes into account the physical and chemical ageing of soot particles (Riemer et al., 2004; Vogel et al., 2009). Values are delivered for the wavelength bands of the radiation scheme used in COSMO-ART (Ritter and Geleyn, 1992). Fundamental input data for the Mie-calculations are the wavelength dependent refractive indices for the individual compounds. Here, we use data of detailed measurements performed in the AIDA (Aerosol Interaction and Dynamics in the Atmosphere) chamber (Schnaiter et al., 2003). The disadvantage of this data is that it was obtained for pure diesel soot. But its advantage is the high spectral resolution of the data which is not the case for other lab studies. A comparison of this fundamental input data with data obtained for biomass burning aerosol is difficult for several reasons. Recent studies ended up with bulk data for mostly aged particles or with mass specific values for extinction and absorption coefficients. Consequently, quite different values were found depending on the specific burning conditions and particle compositions. In many cases values were gained for a single wavelength. For this reason it is difficult to quantify the errors due to the calculation of the optical properties within COSMO-ART.

Following our parameterization we get a value for the mass extinction efficiency of $9.0\,\mathrm{m^2\,g^{-1}}$ for the spectral range $0.25 - 0.7\,\mu m$ and for pure soot particles. For the soot containing Aitken mode we get a value of $5.0\,\mathrm{m^2\,g^{-1}}$, and for the soot containing accumulation mode a value of $4.0\,\mathrm{m^2\,g^{-1}}$. Laser measurements at a wavelength of $0.632\,\mu m$ suggest a value of $7.8\,\mathrm{m^2\,g^{-1}}$ for soot with wood origin (Colbeck et al., 1997).

Levin et al. (2010) carried out measurements with biomass burning aerosol of different chemical composition. The geometric mean diameters ranged from $0.2$ to $0.57\,\mu m$. For those particles they found refractive indices ranging from $1.55$ to $1.80$ for the real part and $0.01 - 0.50$ for the imaginary part. They obtained dry mass extinction efficiency ranging from $1.64$ to $6.64\,\mathrm{m^2\,g^{-1}}$

at a wavelength of $0.532\ \mu\mathrm{m}$. Hungershoefer et al. (2008) found mass extinction efficiencies in the order of $9.0\ \mathrm{m^2\ g^{-1}}$ for savanna grass and African hardwood.

From these numbers we conclude that the optical properties we are using are within the range of literature data.

## 3 Case study

Our simulations are performed for the wildfires that occurred in 2010 in the north of Saskatchewan, Canada. Boreal fires frequently occur following warm and dry periods when the moisture content of the fuel is lowered. A change in the synoptic situation in combination with lightnings and freshening wind often leads to a violent fire condition (Johnson, 1995).

### 3.1 Model configuration

For the fire emissions, a dataset provided by the Global Fire Assimilation System (GFASv1.1) is used. It is a satellite-retrieved
dataset based on daily Fire Radiative Power (FRP) measurements by MODIS on-board the polar orbiting satellites Aqua and Terra. FRP is a quantity that specifies the amount of thermal radiation released by a fire. It is proportional to fuel consumption and emission production (Wooster et al., 2005). Global emission fields for a comprehensive list of species are derived from these FRP observations (Kaiser et al., 2012). Because GFAS uses daily averaged FRP, the retrieved emission rates are also daily products, without any information about diurnal variations. We used 28 gaseous and particulate species derived from
GFASv.1.1, which are allocated into the existing substance classes in COSMO-ART. The species are listed in Table 2 together with their assignments and individual weightings, where necessary.

The simulations are started at 6:00 UTC 10 July 2010, i.e., midnight at local time. The simulations are conducted for 10 days with a horizontal grid spacing of 7 km. The simulation domain is shown in Fig. 3. The fires indicated by the red dots for 15 July 2010 occurred mainly in the north of Saskatchewan. We use 40 vertical non-equidistant layers, and the top of
the model domain is at an altitude of approximately 20 km. The integration time step is 25 seconds. For meteorological initial and boundary conditions, the output of the global model GME is employed (Majewski et al., 2002). The initial and boundary concentrations of trace gases are derived from MOZART-4/GEOS (Wiedinmyer et al., 2011). Biogenic emissions are calculated online within the model system. Furthermore, sea salt emissions are considered because the Hudson Bay is located in the model domain. Anthropogenic emissions are neglected because the model domain is sparsely populated. However,
the province Alberta, located in the western part of the model domain, has oil sand extraction facilities. In comparison with biomass burning emissions, this industry causes only a minor contribution to the aerosol load and is assumed to have no significant influence (Howell et al., 2014).

### 3.2 Simulations

Five different simulations were performed to assess the benefit of a plume rise model. For the simulation VARHEIGHT,
COSMO-ART is applied in combination with the plume rise model mentioned in the previous sections. The parameters fire area and fire intensity have a diurnal cycle according to equation 5. The emission source strength is maintained constant within

24 hours. The simulation EMISSCYCLE is the same, except the emission source strength also has a diurnal cycle. Thus, the emission strength is proportional to the fire size and fire intensity. This means that during night when the fire size and fire intensity are small and the emissions are frequently released at ground level, the emission strength is also low. The simulations 800M and 7500M have the same daily mean emission source strength as VARHEIGHT. However, the emissions are vertically homogeneously distributed from the surface to 800 m and 7500 m, respectively. These distributions are chosen according to Wang et al. (2013) and Pfister et al. (2005), respectively. For the simulation NOFIRE, no biomass burning emissions were considered. All simulations are itemized in Table 3.

## 3.3 Plume heights

Kahn et al. (2007) used MISR data to quantify the physical characteristics of biomass burning plumes. They showed that the biomass burning plumes frequently reach heights well above the planetary boundary layer (PBL) depending on atmospheric stratification. Emissions reaching heights well above the boundary layer encounter reduced vertical mixing. This can lead to well-defined and long-lasting shallow aerosol layers. In this case, it can no longer be assumed that the aerosol above a location of the fire is vertically well mixed. Gonzi et al. (2015) applied a one-dimensional plume rise model in global-scale GEOS-Chem simulations during the year 2006. They concluded that approximately 80 % of the injections of biomass burning aerosol occur below the boundary layer height. Fig. 4 shows the simulated frequency distribution of the simulated plume heights based on the ten-day period of our situation. Thereby every plume top height calculated by the plume rise model is counted. If the fire remains persistent the plume is counted again in the next hour with its new height. Approximately 50 % of the plume heights are below 1 km. At least 30 % of the plume heights are above 2 km and are therefore above the PBL. This percentage represents a lower limit because the PBL height is expected to be quite frequently below 2 km. Ten percent of the plumes are reaching more than 3 km, and the simulated maximum plume height is 8.6 km.

In Fig. 5 the time series of the plume height is depicted for one fire location (56.98° N, 106.99° W). The top blue line denotes the plume top and the lower blue line the plume bottom as simulated by COSMO-ART in combination with the plume rise model. Thereby the upper and the lower bound of the effective emission layer are defined. The diurnal cycle is clearly identifiable. The thickness of the smoke layer depends on the meteorological conditions. During night the smoke is located within 1 km above ground. Daytime values range from about 2 to 7 km for the simulation period. A daily mean is calculated over each day (LST) and averaged over plume top and bottom to obtain a quantity comparable to GFASv1.2 plume height derivations. These are namely the height of maximum injection derived by a later version of the plume rise model within C-IFS (Composition-Integrated Forecasting System) and the plume top estimated after a method by Sofiev et al. (2012). In comparison to the plume heights obtained by simulation VARHEIGHT the GFAS plume heights do not have a diurnal cycle. The daily mean plume height of simulation VARHEIGHT agrees with the GFAS heights in the same extent as they do to each other. According to the GFAS plume height derivations two short fire periods occur during 10. - 19. July 2010 while the fire in simulation VARHEIGHT lasts for ten days.

### 3.4   Comparison with observations

To verify our simulation results and to assess the importance of using a one-dimensional plume rise model, the simulations are compared with different remote sensing products. Note that if errors are made in the estimate of fire intensity and emissions this will influence the concentration in all simulations, while the plume height is only affected in VARHEIGHT and EMISSCYCLE. The satellite image (Fig. 6, left) shows the smoke distribution on 15 July 2010 at 17:55 UTC. The image is in the visible spectral range and was derived by MODIS on-board Terra. The smoke distribution pattern develops in several stages. First, the emissions of the fires are affected by a northern flow from the trough on 11 July 2010 (Fig. 7, left). During the next days, a low-pressure system passes the model domain from west to east, and it is located in the south-west of the model domain on 14 July (Fig. 7, right). In the west of the model domain, the next trough is approaching. The low-pressure system affects the southern fires and transports the smoke to the south. The northern fires are affected by the trough and the corresponding front. Thus, these emissions are transported northwards, and after reaching 64° N, they are bent to the east. The model output of AOD at 550 nm in Fig. 6 on the right is in good agreement with the observed complex smoke structure, e.g. for run VARHEIGHT, which performs best as outlined later. AOD is between 0.1 and 0.5 in areas not affected by smoke. In parts of the dense smoke plume, a maximum of 4 is reached. The comparison shows that the meteorological conditions and the transport processes are well captured with COSMO-ART.

To evaluate the horizontal diffusion of the plume the simulated AOD is compared with AOD satellite retrievals, both at 550 nm. At the top of Fig. 8 observations made by MODIS on-board Terra and retrieved with the dark target algorithm are displayed time averaged over 14 and 15 July 2010. Below the AOD averaged over the four overpass times of Terra satellite are shown for the different simulations. The observed maximum of over 3.5 is located around 57.5° N, 112.5° W. From there the increased AOD is spread towards north-east and south-east. In all simulations the maximum is located further in the east than in the satellite retrieval. The pattern of AOD differs between all simulations in its width, shape, and strength. The southern extension of the plume reaching 50° N, 105° W is best represented by the simulations VARHEIGHT and 800M. Due to the coarse resolution of the satellite retrieval it is not possible to determine the overall best match.

To compare the simulated heights of the smoke layers with observations, retrievals of CALIOP are used. CALIOP on-board the polar orbiting CALIPSO (Cloud-Aerosol Lidar and Infrared Pathfinder Satellite Observation) satellite retrieves vertical profiles of clouds and aerosol along its track (CALIPSO Science Team, 2015). The aerosol products are provided in a horizontal resolution of 5 km x 5 km and 60 m in the vertical direction. The detected aerosol is classified into aerosol subtypes. These different subtypes, namely, clean marine, dust, polluted continental, clean continental, polluted dust, and smoke, are determined with an associated aerosol lidar ratio at 532 nm and 1064 nm (Omar et al., 2009). The confidence in aerosol subtyping can be reduced by thick cloud layers. Furthermore, dense dust and smoke near their source are likely to be misclassified (Liu et al., 2009).

The overpass of CALIPSO at approximately 9:20 UTC on 16 July 2010 is indicated by the purple line in Fig. 3. In Fig. 9a, the CALIOP aerosol retrievals are shown. At the northernmost edge at approximately 64° N, polluted continental aerosol is detected at a height between 6.75 and 9.5 km. Smoke is detected between 63 and 62° N at an altitude from 6 to 7.5 km.

Between 62 and 59° N, the satellite identified smoke, polluted dust and polluted continental aerosol within the lowest 3.5 km. From 56 to 54° N, columns with different heights consisting of smoke, polluted dust, clean continental and polluted continental aerosol are found. Starting at 54° N, a skewed layer of smoke from 5 km down to 3 km at 51° N and further down to 1.5 km at 50° N is observed. Another smoke layer is located between 52 and 51° N at an altitude between 1 and 2 km. Among these smoke regions, polluted dust and, underneath some smoke regions, polluted continental aerosol were detected. The presence of clouds over wide parts of the satellite track lowers the quality of CALIOP aerosol identification. The signal is attenuated. Because aerosol subtypes are pre-defined by distinct values for the extinction-to-backscatter ratio, this could mismatch the actual aerosol type.

A cross-section along the satellite track is conducted for each of the simulations. Smoke is defined to be present when the soot concentration is greater than 0.01 $\mu$g m$^{-3}$. The most prominent features of the observed smoke distribution are marked with dark green circles. Circle A indicates smoke observed by CALIOP between 6 and 7.5 km altitude. This feature is well represented by the simulations 7500M and EMISSCYCLE, moderately represented in VARHEIGHT and 800M fails at this point. Circle B refers to smoke within the lowest 3.5 km. In all simulations the smoke is located a little lower at this position but each of them showing distinct patterns in each case. Circle C and the descending line represent the skewness of the smoke layer between 56 and 50° N. The decline seems to be stronger in the simulations than in the observations. The height is matched by simulation VARHEIGHT and 800M. In EMISSCYCLE the height is slightly overestimated and in 7500M the height is remarkably overestimated. In summary, we found that the simulated smoke distribution depends strongly on the source emission height. For the various scenarios, the occurrence of smoke is very different. When the emission height is restricted to 800 m, the vertical development of the plume is suppressed. Emissions released up to an altitude of 7500 m lead to a further mixing into higher layers. The simulations where the one-dimensional plume rise model is applied perform better.

To further evaluate the individual simulations, we use the AERONET database. An AERONET station is located in the south of our simulation domain. The station Bratts Lake (Fig. 3) provides level 2.0 data for July 2010. In Fig. 10, the AOD is displayed for 15 July 2010 (UTC). The black line denotes the AERONET sun photometer measurements. During night time and under cloudy conditions, no observations are performed. Between 12:00 and 16:00 UTC, the measured AOD reaches 0.6. After a short drop at 16:00 UTC, the AOD increases to 1.1 at 18:00 UTC. The simulation VARHEIGHT agrees very well with the measurements between 12:00 and 15:00 UTC but does not decrease at 16:00 UTC; rather, it reaches the same maximum value one hour earlier. The AOD in the simulation 800M increases to 1.15 at 17:00 and 18:00 UTC. The simulation EMISSCYCLE has its maximum AOD at 16:00 UTC with a value of 1.4. The simulation 7500M does not have that great of a peak; the curve is flat with its maximum of approximately 0.9 at 14:00 UTC. The simulations EMISSCYCLE and VARHEIGHT also show a local maximum at 8:00 UTC with an AOD of 1.2 and 0.9, respectively. In general, COSMO-ART represents the measured AOD well. The best fit is obtained with the simulation VARHEIGHT. The time of the peak AOD is well captured, even though the transport path is quite long. Overall, the simulation VARHEIGHT performs best and is the simulation used in our further investigations to represent the fire case.

In the next section, we show the importance of considering aerosol for numerical weather predictions.

## 3.5 Aerosol radiative impact

The light absorbing and scattering processes of aerosol with exceptionally high concentrations are not considered in most numerical weather prediction models. Instead, an aerosol climatology is used. This leads to erroneous forecasts for temperature and radiation during such events. However, the increasing fraction of solar energy production in many countries requires accurate forecasts of the expected photovoltaic power.

Uncertainties in the radiative impact of biomass burning aerosol are determined by the uncertainties in the description of its optical properties. The optical properties of aerosol depend on the size distribution and the chemical composition. Furthermore, the size distribution and the chemical composition depend in a rather complex way on the fuel type and the mode of combustion (Hosseini et al., 2010). The characterization of the optical properties of aerosol in COSMO-ART is discussed in section 2.4. The simulated number distributions for Fort Smith (60.01° N, 111.57° W; Fig. 3), a location in the fire (61.30° N, 110.45° W), and a location in the vicinity of the fire (58.12° N, 106.51° W) near the surface on 15 July 2010 at 18:00 UTC are shown in Fig. 11. We are not aware of any available in-situ characterizations of the aerosol particles. Instead we compare the model results with the size distribution measured during a small-scale laboratory experiment performed by Hungershoefer et al. (2008). For their experiment savanna grass and African hardwood were burnt in a smoke chamber in order to characterize the optical properties of biomass burning aerosol. At Fort Smith the simulated number concentration is about three orders of magnitude smaller than in the laboratory measurement, while the median diameter is about 0.1 $\mu$m in both cases. At the fire the simulated number concentration is comparable to the measurement but the simulated median diameter of about 0.04 $\mu$m is smaller than in the measurement. Close to the fire the concentration gets smaller and the median diameter bigger than at the fire. The diameter is still smaller and the concentration still higher than in Fort Smith which is located further away from the fire. The aging process can clearly be recognized by the increasing median diameter with distance to the fire. Especially close to the fire the course of measurement and simulation show reasonable agreement. A smaller number concentration can be expected due to dispersion of fresh air outside a laboratory. Sakamoto et al. (2015) specifies the size distribution by a number-median diameter of 0.23 $\mu$m and a standard deviation of 1.7. The values were measured in 1-2-day-old smoke during a campaign in East Canada in 2011. They performed simulations to provide suggestions for the size distribution of fresh emissions. For different entrainment scenarios, they obtained median diameters between 0.059 and 0.094 $\mu$m.

At the top of Fig. 12 the impact on incoming surface short-wave radiation caused by biomass burning aerosol is presented. The figure shows the difference of that quantity between the simulations VARHEIGHT and NOFIRE at the location Fort Smith, which is heavily influenced by smoke, for five days starting on 11 July 2010. On the first day, Fort Smith is not influenced by smoke. Surface short-wave radiation is the same in both scenarios. On 12 July 2010, small differences are simulated, which are mostly caused by changes in cloud cover. The cloud cover is influenced by the biomass burning aerosol; thus, the differences occur in both directions. On the one hand, the availability of aerosol could lead to condensation on these particles and cloud formation; on the other hand, absorbing aerosol could lead to warming and dissolution of clouds. On 13 and 14 July 2010, the incoming surface short-wave radiation reaches approximately 660 W m$^{-2}$ at noon (LT) when the fires are not considered. With biomass burning aerosol, the surface short-wave radiation only reaches a maximum value of approximately 340 W m$^{-2}$.

This means a reduction of up to 50 % due to smoke. On 15 July 2010, a strong decrease also exists, but clouds are also present. The biomass burning aerosol leads to a shift in cloud cover. Consequently, the surface short-wave radiation increases during specific periods. Observations of the global solar radiation at Fort Smith (60.01° N, 111.57° W; Meteomanz.com) do support these simulations. On 15 July 2010 at 6:00 UTC the station reports 1115 J $cm^{-2}$ during the last 24 hours. The simulation VARHEIGHT which includes the fire emissions yields 1029 J $cm^{-2}$ for the same 24 hour period, whilst the simulation

NOFIRE results in 2222 J $cm^{-2}$. This is a typical value for cloudless, smoke-free days. For example on 11 July 2010 a value of 2168 J $cm^{-2}$ was reported at that station.

     Fig. 12 (bottom) illustrates the high sensitivity of radiation forecasts on the parametrization of the plume height and the vertical distribution of the emissions. The various simulations show a very different course of the incoming short-wave radiation for all four days with smoke influence. The altitude and the distribution of the smoke release do not refer to a certain magnitude

of reduction. For example, in the afternoon of 13 July 2010, the simulation 800M yields approximately 520 W $m^{-2}$ and simulation VARHEIGHT only 280 W $m^{-2}$. On the next day, the incoming radiation simulated by 800M is always slightly smaller than that simulated by VARHEIGHT. This implies again the necessity of a correct plume height treatment. As a comparison Bergstrom et al. (2003) calculated a surface radiative forcing of -208 W $m^{-2}$ for Mongu, Zambia, on 6 September 2000 due to biomass burning haze.

Changes in the incoming short-wave radiation at the surface have an impact on the temperature at a height of 2 m. This impact is quantified in Fig. 13 for the entire model domain at 18:00 UTC on 15 July 2012. The temperature reduction is strongest for areas in the north and south of the Great Slave Lake in the north-western part of the model domain. The temperature is reduced by up to 6 K. Most of the areas where a high AOD is simulated in Fig. 6 now belong to a cooling region, with an exception adjacent to the fire location. The lack of a cooling region is due to advection of heated air caused by cloud dissipation

upstream the fires. In the east and in the south-east of Lake Athabasca (approx. 59° N, 110° W), there is an increase in the 2 m temperature, which is caused by changes in cloud patterns. In the remaining model domain, many small areas with a temperature increase alternate with areas with a temperature reduction. The reason for this result is changes in cloud cover due to modified atmospheric flow patterns caused by biomass burning aerosol and perturbed clouds. An even higher temperature difference due to vegetation fires is reported by Ding et al. (2013). During a biomass burning situation in East China, 7.5 K are

in between the forecast and the observations.

     The influence of biomass burning aerosol on the vertical temperature profile is shown in Fig. 14. This figure shows the mean vertical temperature difference between the simulations VARHEIGHT and NOFIRE for a small area around Fort Smith at 18:00 UTC on 15 July 2010. At this time and location, the soot is distributed between the surface and a height of 4 km. The fire aerosol leads to a temperature decrease of up to 4 K at the surface. At a height of approximately 1 km, the sign

changes due to an increase in temperature with a maximum of 1 K at a height of 2 km. At approximately 4.5 km, clearly above the soot layer, the temperature is decreased again, alternating slightly with height up to 15 km. The soot layer absorbs the incoming radiation, leading to a local warming. Less radiation is able to reach the lower levels near the ground, which leads to a temperature reduction in comparison with a smoke-free atmosphere. Temperature changes in various atmospheric layers affect the atmospheric stratification. An increase in static stability is found in such cases. This is in good agreement

with Tummon et al. (2010), who found a decrease in surface turbulent fluxes, a reduced PBL height, and reduced surface temperatures in their climate simulations. Radiative absorption by biomass burning aerosol resulted in diabatic warming of the atmosphere of up to 1 K near 700 hPa. Surface cooling and heating at altitude stabilised the lower troposphere below 700 hPa. Above this, stability was found to be reduced.

## 4 Conclusions

5 We extended the model system COSMO-ART with an online-coupled one-dimensional plume rise model to parametrize the effective source heights for vegetation fires with high energy input. Furthermore, a function to parametrize the diurnal cycle of boreal fires is proposed and included. The improved model system was used to quantify the effects of biomass burning aerosol on radiation and temperature during an intensive fire event that occurred in July 2010 in Canada. Simple parametrizations of the effective source height were compared to the results obtained using the plume rise model. The utilised optical properties and 10 the achieved aerosol size distribution are reasonable and comparable with data from the literature. Comparisons with satellite observations showed that COSMO-ART is able to represent the spatial distribution of the smoke. The simulated AODs with the different assumptions of the effective source height are compared with the observed AODs of the AERONET station Bratts Lake and MODIS AOD retrieval. The simulation where the one-dimensional plume rise model was used together with the diurnal cycle on fire size and intensity match best in magnitude and timing. Including the diurnal cycle of the emissions did 15 not lead to further improvements in our results. This shows that further improvement of the diurnal cycle is required. The vertical extension of the smoke plume was evaluated through a comparison with CALIPSO retrievals. A fixed effective plume height of 7500 m completely overestimates the top of the smoke layer. For the fixed plume height of 800 m, the initial height is frequently exceeded, but elevated smoke layers are not represented. The simulations using the plume rise model performed better. Approximately 50 % of the fire plumes remained in the lowermost 1 km and 30 % of the simulated plumes exceeded a 20 height of 2 km. The fire emissions caused a reduction in surface short-wave downward radiation of up to 50 % under cloudless conditions by absorption in dense smoke layers. The 2 m temperature below these layers decreased by up to 6 K, whereas the temperature in the smoke layer was increased. The temperature change in the column affects the atmospheric stratification. Surface cooling and a warming in elevated layers lead to an increase in atmospheric stability.

*Acknowledgements.* Thanks to I. Abboud and V. Fioletov for their effort in establishing and maintaining the AERONET site Bratts Lake.
25 Thanks to J. Kaiser and S. Remy at ECMWF for providing the GFASv1.1 data set.

We acknowledge the use of Rapid Response imagery from the Land Atmosphere Near-real time Capability for EOS (LANCE) system operated by the NASA/GSFC/Earth Science Data and Information System (ESDIS) with funding provided by NASA/HQ.

The CALIPSO data were obtained from the NASA Langley Research Center Atmospheric Science Data Center.

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

**Table 1.** Chemical composition, initial number-median diameter, and standard deviation of the lognormal distributions of the eight modes. (AIT: Aitken mode, AITS: Aitken mode containing a soot core, ACC: accumulation mode, ACCS: accumulation mode containing a soot core, SOOT: pure soot mode, SEASA: sea salt fine, SEASB: sea salt medium, SEASC: sea salt coarse)

| | AIT | AITS | ACC | ACCS | SOOT | SEASA | SEASB | SEASC |
|---|---|---|---|---|---|---|---|---|
| soot | | ● | | ● | ● | | | |
| sulfate | ● | ● | ● | ● | | ● | ● | ● |
| ammonium | ● | ● | ● | ● | | | | |
| nitrate | ● | ● | ● | ● | | | | |
| organics | ● | ● | ● | ● | | | | |
| water | ● | ● | ● | ● | | ● | ● | ● |
| sodium chloride | | | | | | ● | ● | ● |
| Initial diameter in $\mu$m | | | | | | | | |
| | 0.01 | 0.08 | 0.07 | 0.08 | 0.08 | 0.2 | 2 | 12 |
| Standard deviation | | | | | | | | |
| | 1.7 | 1.7 | 2.0 | 2.0 | 1.4 | 1.9 | 2.0 | 1.7 |

**Table 2.** Emitted gaseous and particulate species derived from GFASv1.1

| original notation | COSMO-ART class |
| --- | --- |
| Carbon Monoxide | CO |
| Nitrogen Oxides $NO_x * 0.9$ | NO |
| Nitrogen Oxides $NO_x * 0.1$ | NO2 |
| Sulfur Dioxide | SO2 |
| Ammonia ($NH_3$) | NH3 |
| Ethane ($C_2H_6$) | ETH (Ethan) |
| Methanol ($CH_3OH$) | HC3 ($C_3$ to $C_5$ Alkanes) |
| Ethanol ($C_2H_5OH$) | HC3 ($C_3$ to $C_5$ Alkanes) |
| Propane ($C_3H_8$) | HC3 ($C_3$ to $C_5$ Alkanes) |
| Butanes ($C_4H_{10}$) | HC3 ($C_3$ to $C_5$ Alkanes) |
| Pentanes ($C_5H_{12}$) | HC5 ($C_6$ to $C_8$ Alkanes) |
| Hexanes ($C_6H_{14}$) | HC5 ($C_6$ to $C_8$ Alkanes) |
| Heptane ($C_7H_{16}$) | HC8 (higher Alkanes) |
| Ethene ($C_2H_4$) | OL2 (Ethene) |
| Propene ($C_3H_6$) | OLT (terminal Alkenes) |
| Butenes ($C_4H_8$) | OLT (terminal Alkenes) |
| Octene ($C_8H_{16}$) | OLT (terminal Alkenes) |
| Pentenes ($C_5H_{10}$) | OLI (internal Alkenes) |
| Hexene ($C_6H_{12}$) | OLI (internal Alkenes) |
| Isoprene ($C_5H_8$) | ISO (Isoprene) |
| Terpenes ($C_5H_8$)n | API (Terpenes) |
| Toluene ($C_7H_8$) | TOL (Toluene) |
| Benzene ($C_6H_6$) | TOL (Toluene) |
| Xylene ($C_8H_{10}$) | XYL (Xylene) |
| Formaldehyde ($CH_2O$) | HCHO |
| Acetaldehyde ($C_2H_4O$) | ALD (Acetaldehyde) |
| Acetone ($C_3H_6O$) | KET (Ketones) |
| Black Carbon | s (pure soot mode) |
| Organic Carbon $* 0.1$ | if (Aitken mode particles, soot free) |
| Organic Carbon $* 0.9$ | jf (Accummulation mode particles, soot free) |

**Table 3.** Specification of the realised simulations

| | diurnal cycle fire intensity & size | diurnal cycle emission strength | vert. dist. profile | homogeneous vert. dist. |
|---|---|---|---|---|
| VARHEIGHT | ● | | ● | |
| EMISSCYCLE | ● | ● | ● | |
| 800M | | | | ● |
| 7500M | | | | ● |
| NOFIRE | | | | |

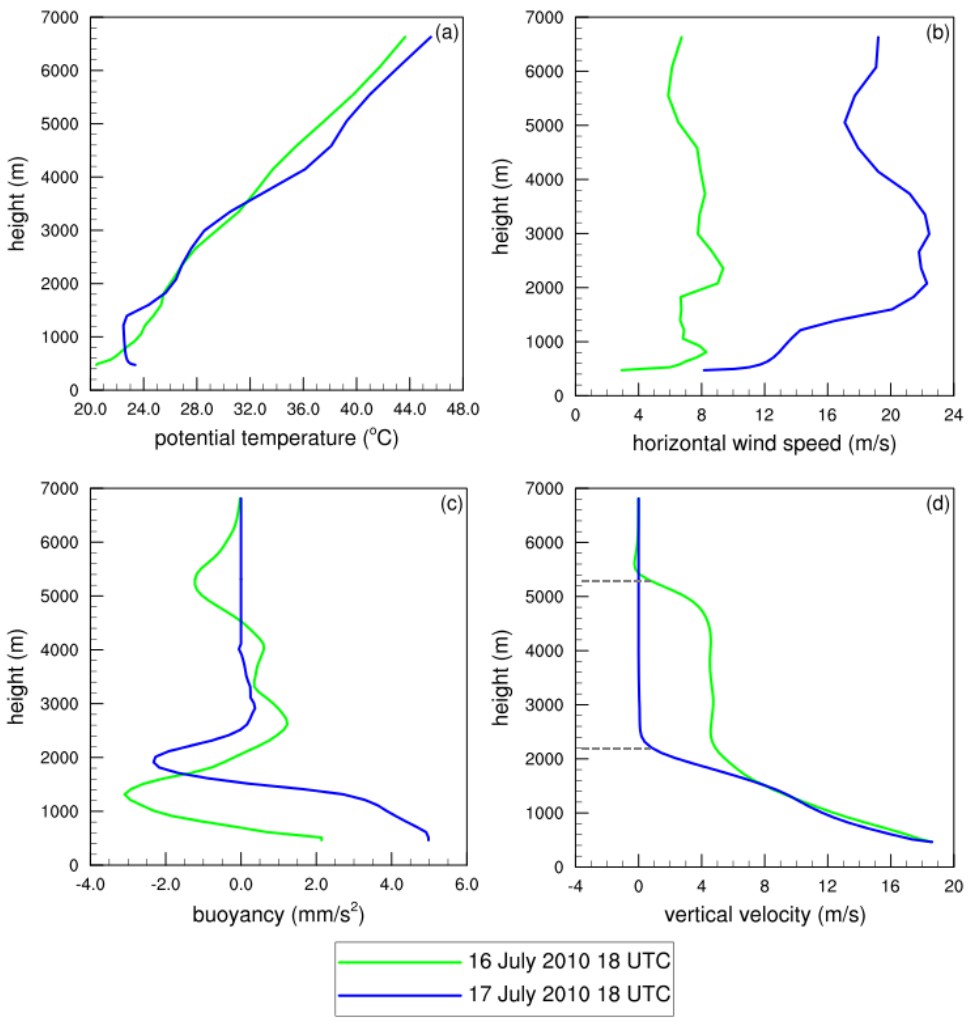

**Figure 1.** The potential temperature (a) and horizontal wind speed, (b) of the environment and the calculated buoyancy, and (c) and vertical velocity (d) for an imaginary fire in central Saskatchewan, Canada, for 16 July 2010 at 18:00 UTC and 17 July 2010 at 18:00 UTC.

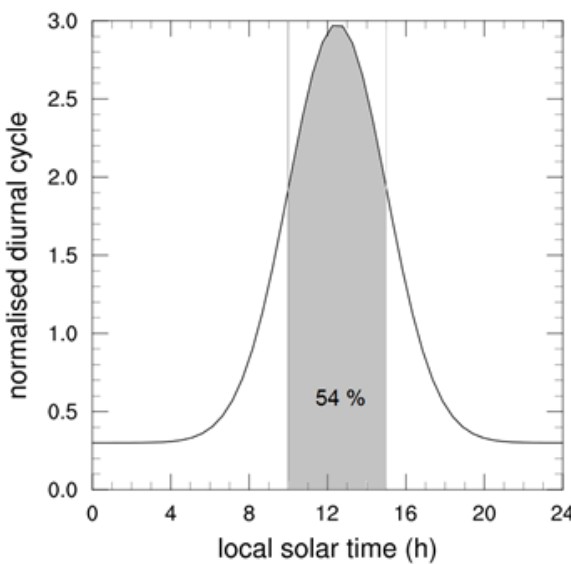

**Figure 2.** The course of the diurnal cycle assumed for fires in boreal forests. In the individual simulations this diurnal cycle is overlaid on the daily values of fire size, fire intensity and emission strength.

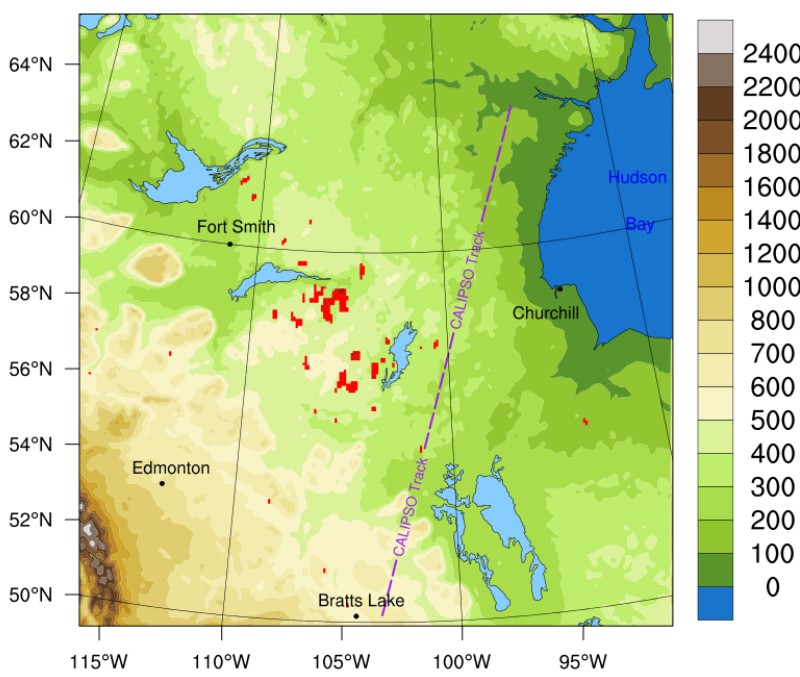

**Figure 3.** Simulation domain with model orography without the relaxation area. The fire locations are marked in red for 15 July 2014. The CALIPSO overpass at around 9:20 UTC 16 July 2010 is denoted by a purple line.

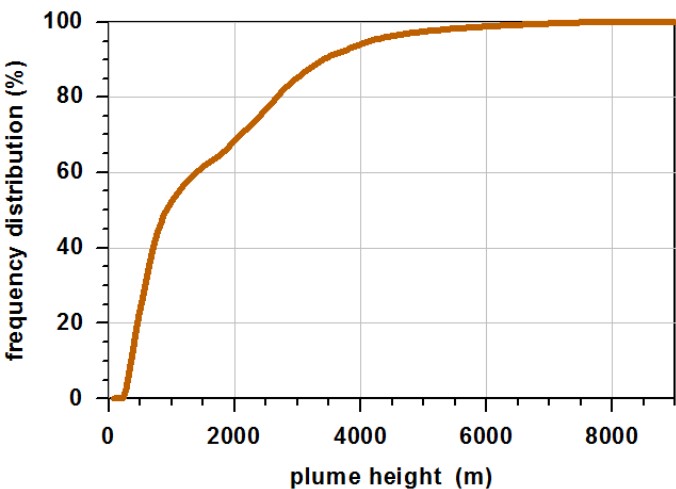

**Figure 4.** Accumulated frequency distribution of the simulated plume heights for the ten day time period from 10 July 2010 to 20 July 2010. In total 52628 data points are included.

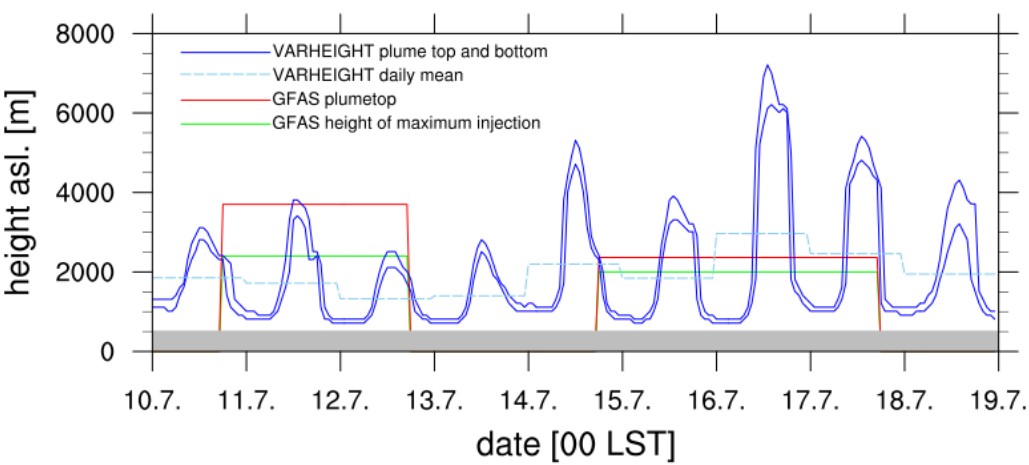

**Figure 5.** Time series of the plume height at one fire location. The lower and the upper bound of the effective emission layer simulated by the plume rise model within COSMO-ART (blue lines). The dashed light blue line gives the daily mean of these heights. Plume heights specified by GFASv1.2 for two derivations (red and green line).

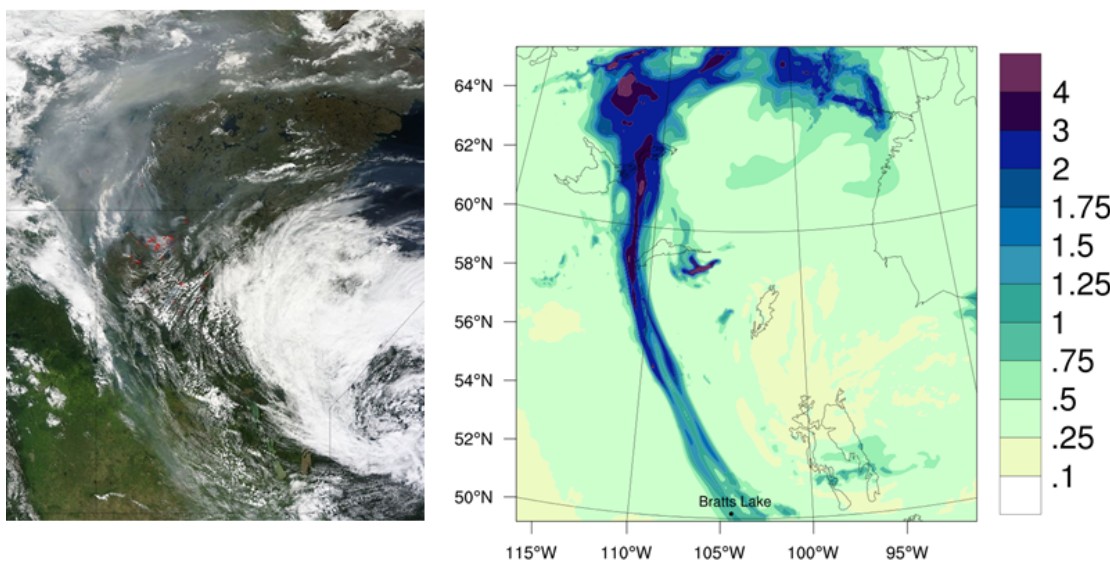

**Figure 6.** Left: Satellite image in the visible section at 17:55 UTC 15 July 2010. The center of the image is located at 56° N and 102° E. The red dots denote the fire locations. The grey structures state the distribution of smoke (LANCE Rapid Response MODIS Image Gallery, NASA). Right: Simulated aerosol optical depth at 18:00 UTC 15 July 2010.

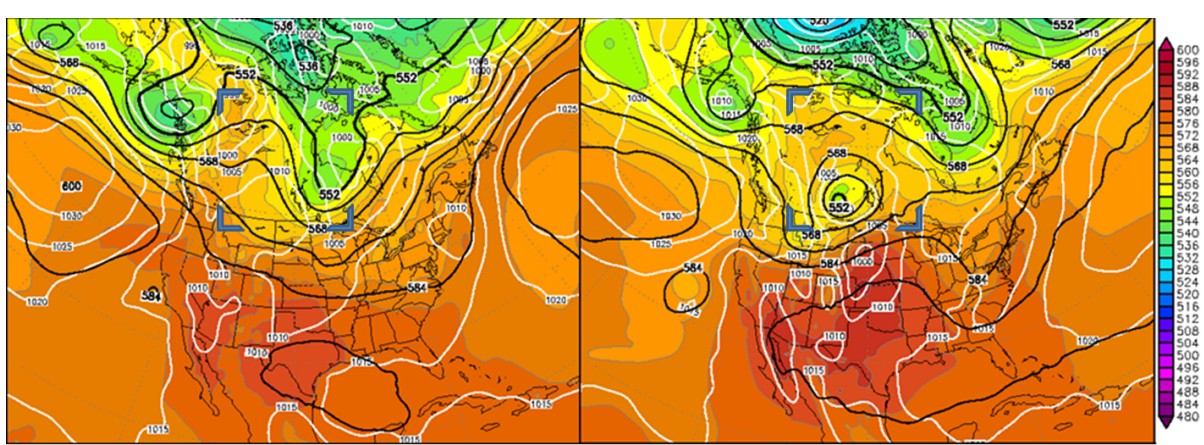

**Figure 7.** Reanalysis (Climate Forecast System, CFS) of the meteorological conditions on 11 July 2010 at 18:00 UTC (left) and 14 July 2010 at 6:00 UTC (right), 500 hPa geopotential (gpdm) black lines, surface pressure (hPa) white lines and relative topography H500-H1000 (gpdm) colour coding (www.wetter3.de). The edges of the simulation domain are indicated with blue triangles.

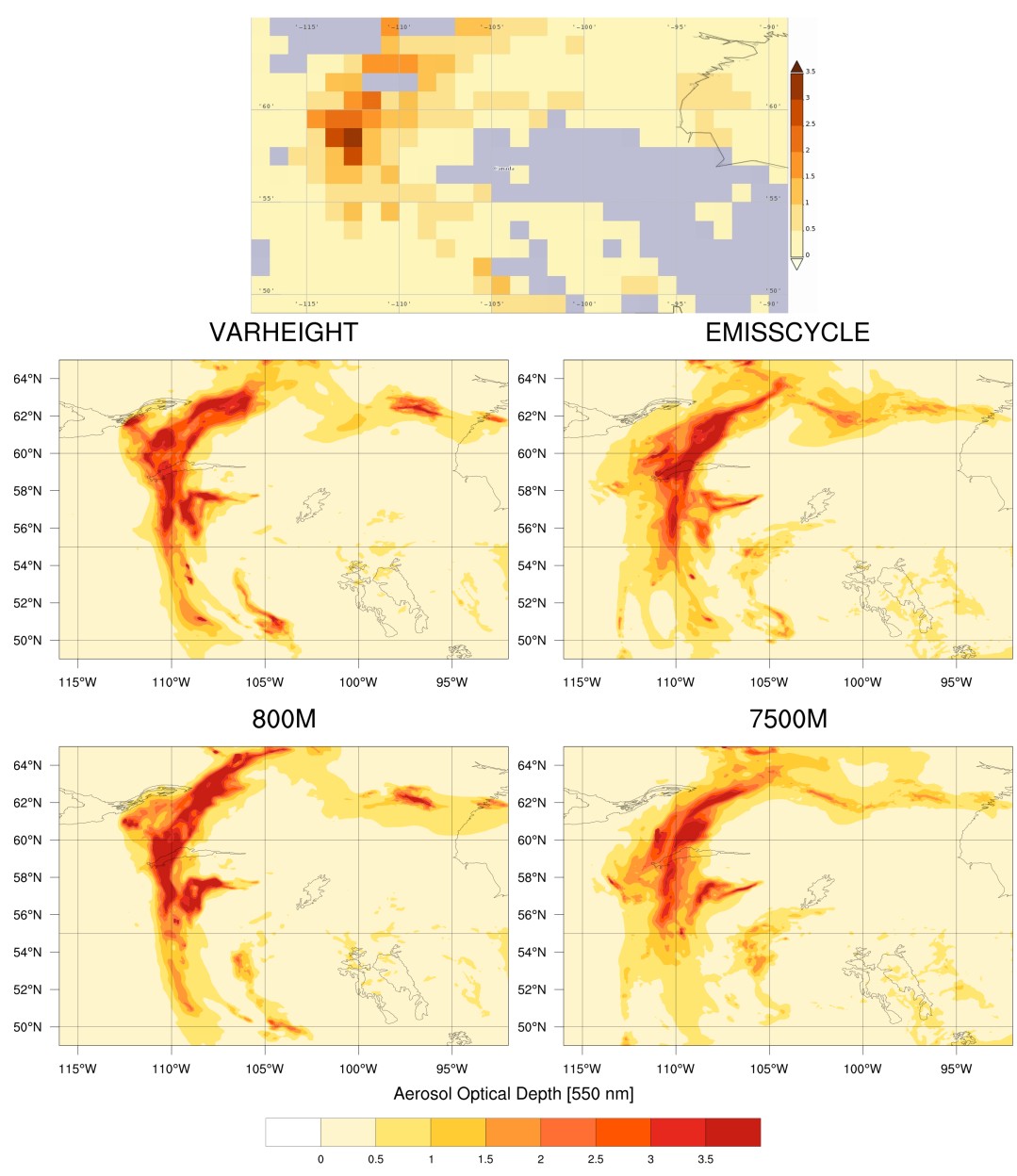

**Figure 8.** AOD at 550 nm averaged over 14-15 July 2010. Top: Satellite retrieval from MODIS on-board Terra, below: Simulations VARHEIGHT, EMISSIONCYCLE, 800M, and 7500M.

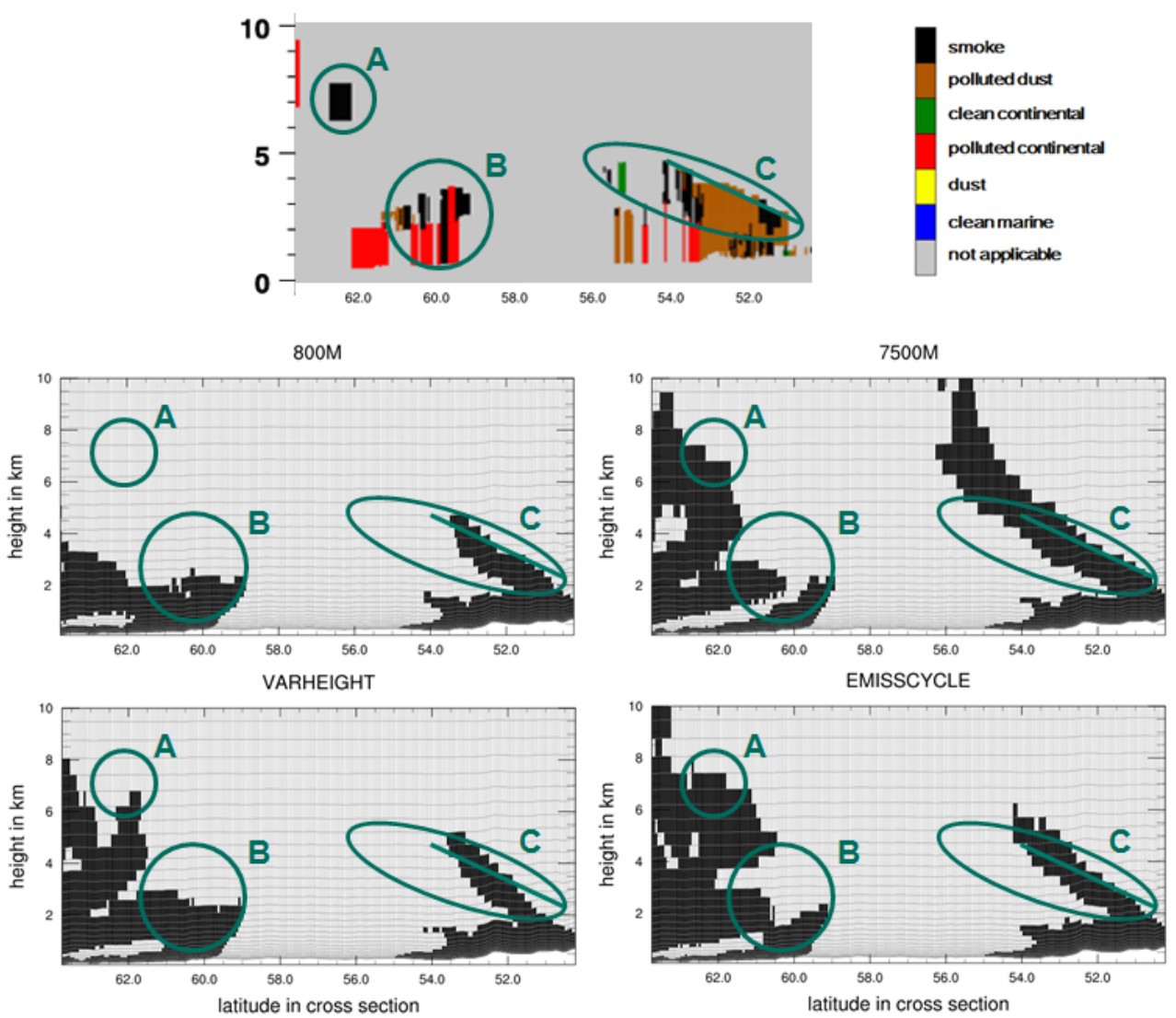

**Figure 9.** Cross section of aerosol subtypes of CALIPSO overpass at around 9:20 UTC 16 July 2010 (a). The black colour coding denotes the presence of smoke; brown, green, and red represents polluted dust, clean continental, and polluted continental, respectively. Cross section along the same CALIPSO track for simulations 800M (b), 7500M (c), VARHEIGHT (d) and EMISSCYCLE (e), here only soot concentrations greater than 0.01 $\mu$g m$^{-3}$ are displayed.

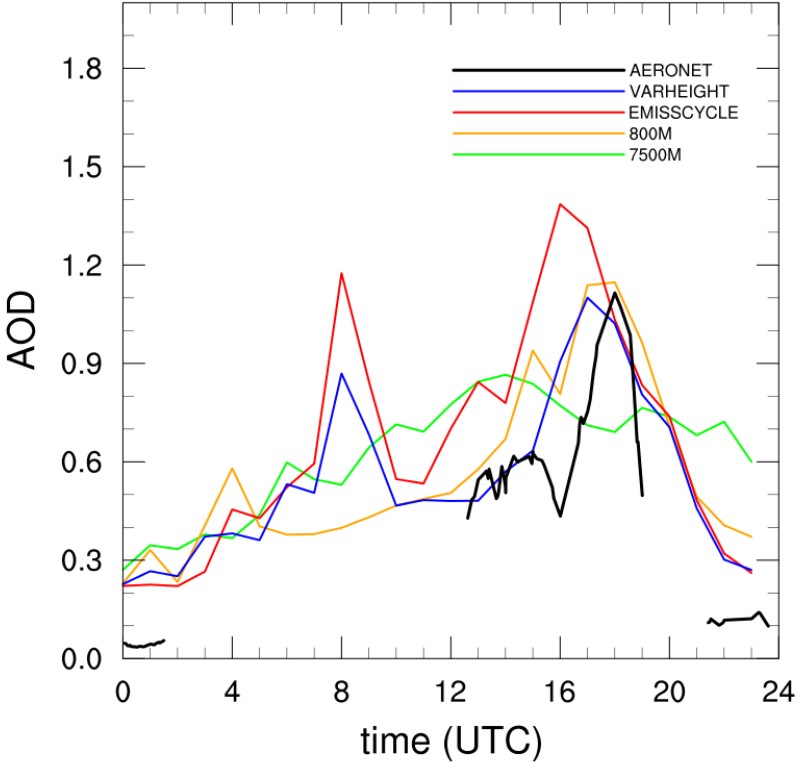

**Figure 10.** Comparison of measured (AERONET) and simulated AOD at Bratts Lake for 15 July 2010.

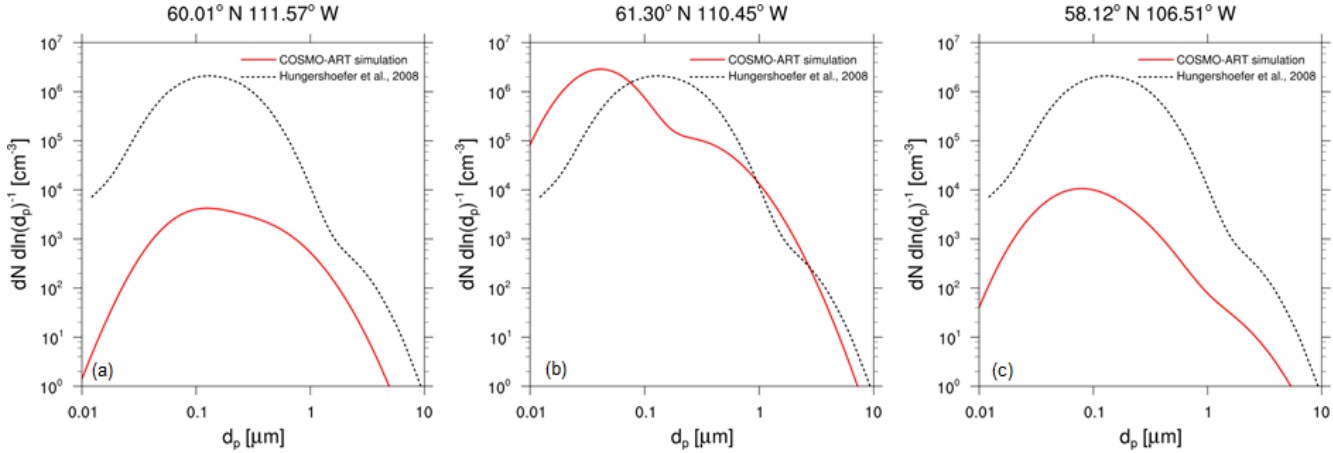

**Figure 11.** Number size distribution (a) at 60.01° N, 111.57° W (Fort Smith), (b) 61.30° N, 110.45° W (at fire), and (c) 58.12° N, 106.51° W (close to fire) near the surface on 15 July 2010 at 18:00 UTC for VARHEIGHT (red line) and as comparison measurements from an experimental fire (doted black line) performed by Hungershoefer et al. (2008).

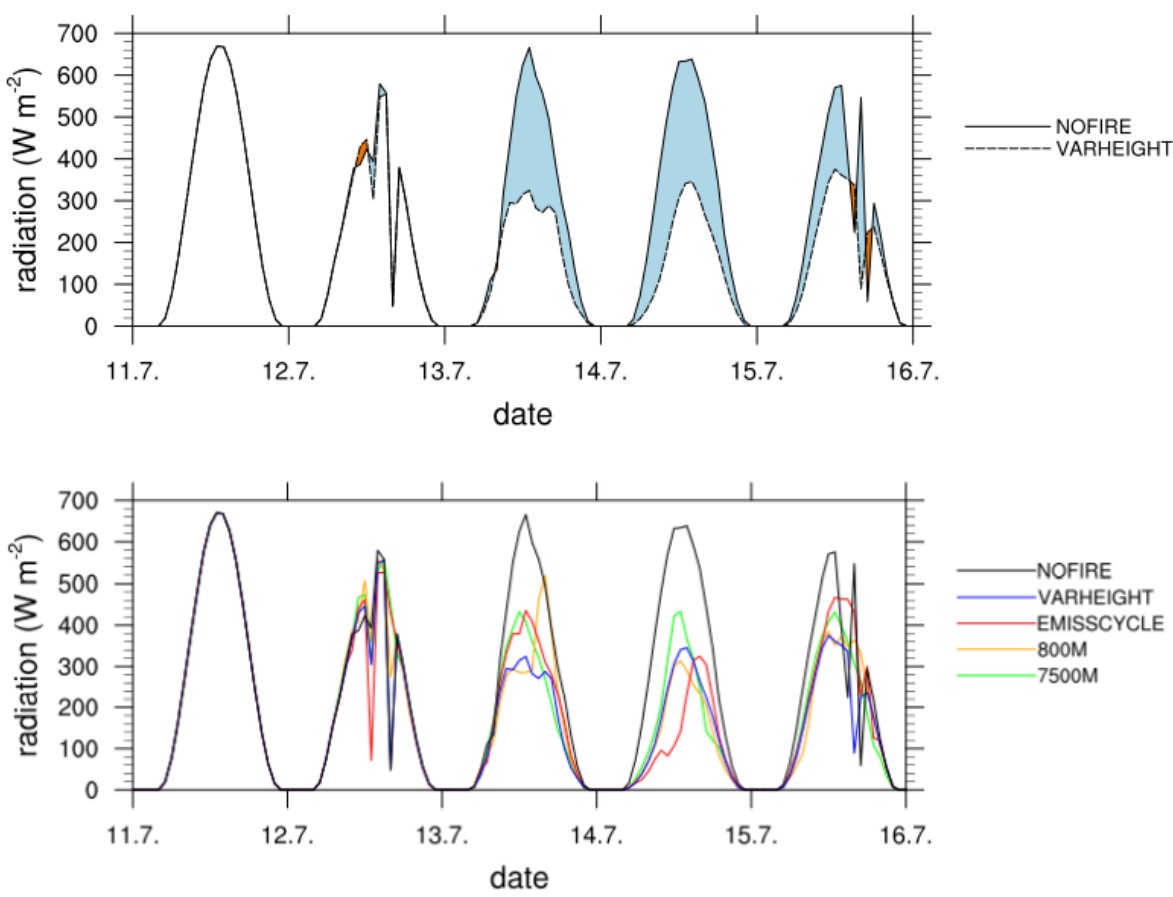

**Figure 12.** Surface short-wave radiation in Fort Smith for five days starting on 11 July 2010, for the simulation without fire emissions and a simulation with fire emissions (top), and the same for the simulations with different parametrisations for plume height and vertical distribution of the emissions (bottom).

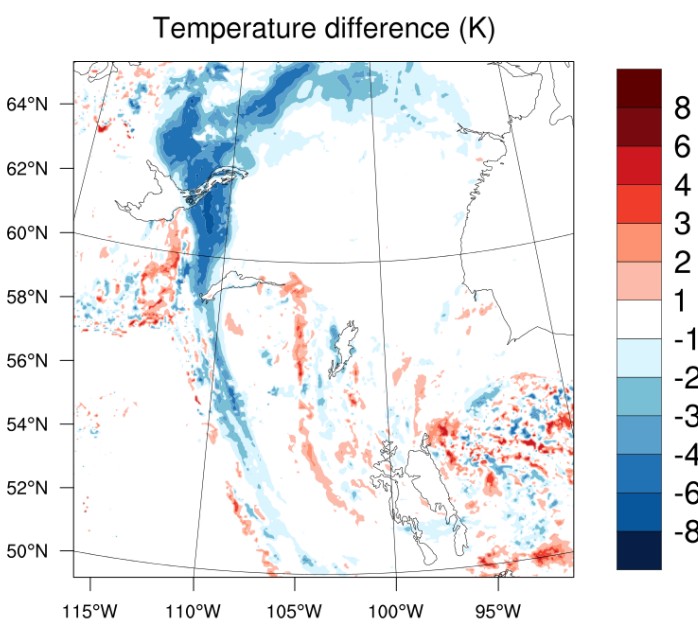

**Figure 13.** Temperature changes in 2 m height at 18:00 UTC 15 July 2010. Displayed is the difference between the simulations VARHEIGHT and NOFIRE.

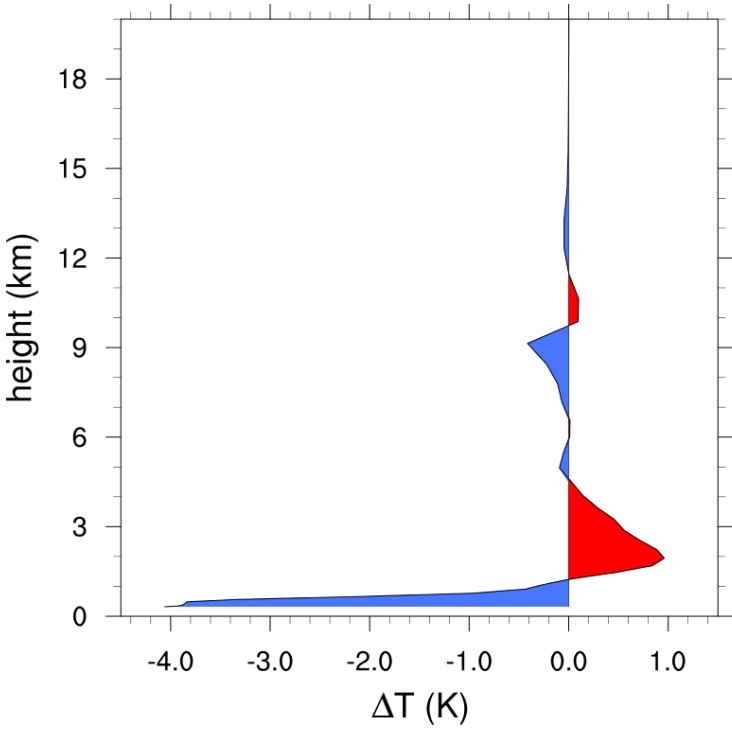

**Figure 14.** Mean vertical temperature change between the simulations VARHEIGHT and NOFIRE for a small domain around Fort Smith for 18:00 UTC 15 July 2010.