# Peer review of "The Importance of Plume Rise on the Concentrations and Atmospheric Impacts of Biomass Burning Aerosol"

_Atmospheric Chemistry and Physics, 2015_

## Referee Comment (RC1) · Anonymous Referee #2 · 16 Feb 2016

Referee report for ACP-2015-964 « The Importance of Plume Rise on the Concentrations and Atmospheric Impacts of Biomass Burning Aerosol"

The authors describe how a 1D plume rise model was integrated into the COSMO- ART forecasting system. The extended COSMO-ART was then used, with different assumptions about the vertical distribution of aerosol emisisons, to simulate an intense fire in Saskatchewan in July 2010.

The scientific significance of this work is great, since the injection heights of biomass burning emissions have been shown to be a key source of uncertainty in atmospheric composition forecasts. The implementation of the Freitas plume rise model in COSMO-ART is a notable extension of this system, and this extension was well used in this case study. This case study however doesn't amount to a full validation of the COSMO-ART-PRM system: it would be nice if the authors discuss global comparison against observations such as the MPHP or MPHP2 datasets for an extended period of time (one year or more). If this was not done, then maybe it could be mentioned in the perspective section.

The authors show a good command of the COSMO-ART itself, and of the issues around biomass burning aerosols. The case study is thoroughly investigated in terms of radiative and meteorological impacts. The article gives more emphasis to the impact of the various simulations on the vertical distribution of biomass burning aerosols, compared to remote observations. The impact on the horizontal diffusion of the plume (as compared to MODIS AOD observations, for example) has been less studied: the authors could maybe show a plot to describe this aspect.

This paper allows for a reduction of the uncertainties in biomass burning aerosol forecasts. However, the other sources of uncertainties (turbulent diffusion, transport), etc...have not really been mentioned. While it is a hard job to estimate these sources of error, maybe a comparison of the forecasted meteorological parameters against observations (weather stations, reanalysis or radio-soundings if any radio-sounding is available in this area) could help.

The presentation of the paper is very good and the standard of English excellent. The plots are well chosen and very clear. The new plots 9 and 10 are more readable than before. Maybe plot 13 could also be enlarge as well.

Overall, this is an excellent article and I recommend it to be published with minor corrections.

Specific comments :
- Page 3 line 5 "Additional buoyancy can be gained through release of latent heat": for large fires latent heat can be an important contribution (pyro-CU and Cb)

- Page 3 lines 10-25: see the review of Paugam et al 2015 : Paugam, R., Wooster, M., Freitas, S., and Val Martin, M.: A review of approaches to estimate wildfire plume injection height within large-scale atmospheric chemical transport models, Atmos. Chem. Phys., 16, 907-925, doi:10.5194/acp-16-907-2016, 2016.

- Page 5 line 19 "To demonstrate the importance of meteorological conditions on the maximum height of the plume top": indeed, sometimes the meteorological conditions can have more impact on the plume top height than the fire itself. In our experience with a later version of Freitas's PRM, the values for median injection height were sometimes higher with no fire forcing at the base than with fire forcing, which is anomalous (this happened in around 10% of cases with Aqua/Terra pixels). The authors are encouraged to test this kind of occurrence.

- Page 6: Since Freitas's PRM provides a detrainment profile, I don't understand why the vertical distribution of emissions has to be parameterized in such a way. Instead of getting just the lower and upper bounds from the PRM, isn't it possible to get the whole detrainment profile and then interpolate it to COSMO-ART levels? Otherwise, the proposed parameterization seems sensible.

- Page 7, diurnal cycle section: the approach is alright. On this subject you can also refer to Andela, N., Kaiser, J. W., van der Werf, G. R., and Wooster, M. J.: New fire diurnal cycle characterizations to improve fire radiative energy assessments made from MODIS observations, Atmos. Chem. Phys., 15, 8831-8846, doi:10.5194/acp-15-8831-2015, 2015.

- Page 8, Model configuration: There is a new GFAS dataset, GFASv1.2, which includes "mean heights of maximum injection" (the average of the PRM levels where detrainment is above half of the maximum detrainment) and "plume top", computed by the PRM from Freitas (and updated by R. Paugam) using MODIS observations and ECMWF meteorological profiles. It also includes injection heights computed following Sofiev et al. (2012). It would be interesting for you to compare these data with the plume top that you obtained with the PRM.
- Page 9 Plume heights: Since it is the main subject of the paper, it would be nice to have more information on the plume heights provided by COSMO-ART-PRM, maybe the top and bottom of the plume at some selected locations and times for example, or the emission profiles used in COSMO-ART.

---

## Referee Comment (RC2) · Anonymous Referee #3 · 1 Mar 2016

GENERAL REMARKS The paper presents interesting results on how different treatments of plume rise of fire emissions impact on aerosol concentrations, and radiative impact. The paper shows the significant impact of fire emissions on the atmosphere's radiative budget. The paper would be worthwhile for publication in ACP if the following major remarks were properly taken into account, in particular: • a sound argumentation on the choice of several parameters (in particular fire intensity), • a better description on how optical properties and aerosol-cloud interactions are calculated in the model.

MAJOR REMARKS: Section 2.1, page 5, lines 13 – 20. The parameters used in this study to obtain the lower and upper bounds of the plume height need to be much

better justified, in the context of available fire studies. When reading the paper, one could think that the two upper and lower values given for fire intensity (30 and 80 kw/m2) represent a common range of observed values. But then these values are not really used n a statistical sense, but rather in a deterministic way to calculate lower and upper plume heights for a given fire. Isn't there a conceptual mismatch. Also the values chosen for the limiting vertical velocity and default fire size need justification. For all values, how would altering them with respect to their estimated uncertainty ranges alter the results of this paper. Some sensitivity tests would be welcome here.

Section 2.4, page 7: This section is difficult to read, because the aim of the argumentation is not clear from the beginning on. The last sentence, that the authors were unable to perform Mie calculations for this study, and thus took values for diesel soot instead of wood soot should be put right in the beginning of the section. Potential implications of this approximation should be discussed all along the paper, in particular in section 3.5 (radiative effects). Fire aerosol is also constituted of organic aerosol. Which optical properties are adopted for organic aerosol ? Is internal or external mixing assumed for different fire aerosol components ? This should be stated. Only one reference for one wavelength is given for the single scattering albedo of diesel and wood. I guess that there are much more results available in literature. Please synthesize. Optical parameters of soot have been shown to change with plume age (for example review of Bond et al., 2013). This effect is not considered in the present study. This point should at least be discussed. Please also discuss, how specific information on size distribution would ideally be used for Mie calculations, and how this was handled in the present study. Again, what is the expected error ?

Additional section 2.5: Please describe, how aerosol microphysics interactions are treated in the model, which processes and parameterizations are included? This is crucial for enabling the reader to understand results presented in Section 3.5 (Aerosol radiative impact).

Section 3.4 : The arguments given for stating that the VARHEIGHT simulation is the

best are to some extent convincing. Nevertheless, the given data set is quite restricted, are there more observations available ? For instance in-situ PM measurements at surface sites? MODIS or POLDER AOD fields ? Is it possible to put the discussion on a more quantitative basis (for example by calculation of correlation coefficients between simulations and observations? It should be mentioned while discussing results in section 3.4, that differences between simulations and observations could be due also to errors in fire intensity and emissions. In how far do such errors prohibit from drawing conclusions on the different plume rise schemes. Overall, section 3.4 is quite difficult to follow, may be it is possible to simplify, and not give all numbers. Those could be grouped together in a table.

Section 3.5 would be strengthened, if simulated effects on short wave radiation, temperature and cloud cover could be substantiated by observations, for the given case study. This should be possible from meteorological in situ and satellite observations. Without observations, this section remains rather speculative.

MINOR REMARKS : Page 3, lines 7-9: Is this rapid transport to Europe due to prior vertical lifting into the upper troposphere with stronger winds. Please make this link clear in the revised text. Page 3, lines 19-23: are these arguments valid for specific cases or are they more general, please make this clear. Page 4, lines 11-19: please better argue, why this study is new with respect to older work . Page 4, model description: Is secondary aerosol formation from biomass burning emissions included in the model ? This process is for example shown to be important for Russian fires in summer 2010 (Konovalov et al., 2015). Page 5, Section 2.1: is lateral detrainment in the convective fire plume is apparently not considered ? Page 9, line 18: 'in sufficient agreement' agreement with what ? Page 11, line 32: A median mass diameter above $1\mu g/m3$ seems large to me. It is for instance larger than the accumulation mode in which most mass of continental aged pollution aerosol is concentrated. Is there an explanation, why this is different for fire aerosol.

Tables : It would be worthwhile to add a table with emission factors for different model

species.

Figures :

Figure 2: please specify the Figure legend, for which parameter the diurnal cycle is shown ?

Figure 4: What is the meaning of the red points ? I guess the smoke area is in grey, while clouds are white. To be completely clear, this could be mentioned in the legend. How does observed smoke region compare to that simulated with different plume height options. Does such a comparison allow state on benefits of different plume height treatments?

Figure 5: It is difficult to make the "geographical" link between figures 4 and 5. In figure 5, please indicate latitudes and longitudes, or make appear the domain of fig. 4 in fig. 5.

Figure 7: How are colors attributed, it is not very quantitative ? This is mentioned in the main text, please recall it in the figure legend.

Figure 10: Aren't there any observations of short wave radiation available in the modelling domain? Is figure 10 contained in Figure 13, or is it different. It could be justified, but please indicate it.

TECHNICAL, EDITING REMARKS : Page 2, line 7: 'they developped' -> 'Konovalov et al.' or 'the authors' Page 3, line 18: 'Another simulation . . ..' In the same study/reference ? Page 3, line 18: 'the same' Which ? Page 6, line 18: 'the wind speed in the boundary layer is usually higher . . ..' Add 'usually'

REFERENCES :

Bond, T. C., Doherty, S. J., Fahey, D. W., Forster, P. M., Berntsen, T., DeAngelo, B. J., Flanner, M. G., Ghan, S., Kärcher, B., Koch, D., Kinne, S., Kondo, Y., Quinn, P. K., Sarofim, M. C., Schultz, M. G., Schulz, M., Venkataraman, C., Zhang, H., Zhang, S.,

Bellouin, N., Guttikunda, S. K., Hopke, P. K., Jacobson, M. Z., Kaiser, J. W., Klimont, Z., Lohmann, U., Schwarz, J. P., Shindell, D., Storelvmo, T., Warren, S. G., and Zender, C. S. : Bounding the role of black carbon in the climate system : A scientiïfic assessment, J. Geophys. Res.-Atmos., 118, 1–173, 2013.

Konovalov, I. B., Beekmann, M., Berezin, E. V., Petetin, H., Mielonen, T., Kuznetsova, I. N., and Andreae, M. O.: The role of semi-volatile organic compounds in the mesoscale evolution of biomass burning aerosol: a modeling case study of the 2010 mega-fire event in Russia, Atmos. Chem. Phys., 15, 13269-13297, doi:10.5194/acp-15-13269-2015, 2015.

---

## Referee Comment (RC3) · Anonymous Referee #1 · 4 Mar 2016

The authors present the implementation and the application of a documented biomass burning plume rise model in the COSMO-ART regional model, a state-of-the-art meteorological and chemical transport model. The integration of the plume rise model is appropriate for the COSMO-ART modeling system and its inclusion clearly improves the physical description of biomass burning emissions from wildland fires in the model. On the other hand, the plume rise model does contain a set of parameters, e.g., fire size, definition of the plume outflow levels, which still need to be pre-selected. The authors present the results of multiple model simulations from a 10-day period in July 2010 using different assumptions on the height of the emitted biomass burning emission in Canada. They conclude that the application of the plume rise model improves

the model simulations in comparison with satellite and surface observations.

The paper well written; it provides an appropriate review of the available literature and contains a fair amount of detail on the COSMO-ART and the plume rise model. The selected scenario, i.e., the 2010 wildland fires in Canada, fits the purpose to assess the impact of the model performance by including the plume rise model. I recommend to accepted this manuscript for publication in ACP after my minor comments are addressed.

Specific Comments:

- Line 36: Please replace 'they' with 'Konovalov et al., 2015' to clarify the reference.

- Line 136: Please add a reference for the used data set for the fire emissions; I guess Kaiser et al., 2012, would be appropriate when referring to GFAS, but maybe there is a more recent reference available.

- Section 2.3: The time step of the plume model calculations is not clearly stated in this section; I assume the emissions are updated with an hourly time step, but it would be useful to have this stated here (or in Section 3.1 in case the time step is flexible and can be easily adjusted according to the model simulation, similar to the integration time step).

- Section 2.4: The chemical composition and the optical properties of biomass burning aerosol are rather complex; to the best of my knowledge it is still uncertain whether fuel load and/or fire type (i.e, smoldering or flaming) determine the optical properties of the biomass burning aerosol. Hence, certain assumptions on the optical properties of the emitted aerosol have to be made in model simulations such as those presented here. However, please refer to and discuss some references dealing with the study of biomass burning aerosol and its optical properties, e.g., Hungershöfer et al., 2008; Levin et al., 2010; Saleh et al., 2014.

- Line 257 ff: Clearly the use of the single-scattering albedo for diesel soot results in an

overestimation of the absorption of the emitted wildland fire aerosol, as correctly stated in the manuscript. Since the improved treatment of the optical properties of the emitted aerosol is not the main purpose of this study, it seems appropriate for the present work to use the aerosol classes available in the modeling system. However, for follow-up studies, in particular studies related to the dynamical feedback of the biomass burning aerosol on the atmosphere through aerosol absorption, this significant limitation of the model systems requires improvement. For the current study, please remove 'may slightly' from the final sentence of this paragraph so that it reads: 'Using the optical properties of diesel soot for our simulations, we overestimate the absorption in layers of dense smoke.'

- Line 272: Please check whether the reference to Kaiser et al.., 2009a, can be replaced by referring to Kaiser et al., 2012, which is a peer-reviewed publication and not a Technical Document.

- Line 275: Please add some more information on the properties of the emitted aerosol particles; e.g., to which modes and composition the emitted aerosol particles are allocated. These classes could maybe be highlighted in Table 1.

- Line 311: What is the frequency of the plume height calculation used to generate Figure 6? Does the plume height represent the hourly emission height (i.e., every fire plume being counted multiple times) or the mean for each fire over a certain period (i.e., every fire counted only once). Please specify.

- Line 324: Please replace 'through' by 'trough'

- Line 353: Please start a new paragraph after '. . .aerosol type.'.

- Line 354 – 382: This paragraph is rather hard to follow; from my perspective it contains too many numbers. The authors might consider to add a table with the corresponding numbers and to substantially shorten this section.

- Line 390 ff: The comparison with the data from the AERONET station at Bratts Lake

is only performed for a single day (15 July). Would it be possible to repeat this analysis for other days, in particular for 16 July when the CALIPSO data are available. Please extend the comparison with available AERONET data from other days in July 2010.

- Section 3.5: Please clearly state at the beginning of this section the limitation of the analysis of the radiative impact of the biomass burning aerosol due to the use of the optical properties from diesel soot instead of biomass burning aerosol.

- Line 430 ff: Please motivate the use of Fort Smith to assess the aerosol impact on surface solar radiation. Obviously it would be very valuable if surface measurements would be available to complement the comparison between the different model simulations. Are there corresponding measurements available at the AERONET site in Bratts Lake?

- Figure 11: It is striking that no temperature change is simulated around 106°W / 58°N, despite the high aerosol loading as shown in Figure 4. Please comment.

- Line 475 / Figure 13: Move this paragraph and the figure towards Fig. 10 and the corresponding text.

References

Hungershöfer et al., (2008), Modelling the optical properties of fresh biomass burning aerosol produced in a smoke chamber: results from the EFEU campaign, Atmos. Chem. Phys., 8, 3427–3439, 2008.

Kaiser, J. W., et al. (2012), Biomass burning emissions estimated with a global fire assimilation system based on observed fire radiative power, Biogeosciences, 9(1), 527-554, doi:10.5194/bg-9-527-2012.

Levin, E. J. T., et al. (2010), Biomass burning smoke aerosol properties measured during Fire Laboratory at Missoula Experiments (FLAME), J. Geophys. Res., 115, D18210, doi:10.1029/2009JD013601.

Saleh, R., et al. (2014), Brownness of organics in aerosols from biomass burning linked to their black carbon content, Nature Geoscience, 7(9), 647-650, doi:10.1038/ngeo2220.

---

## Author Comment (AC1) · 31 May 2016

We thank referee #2 for his valuable comments and suggestions. We followed them as explained below.

The reviewers comments are repeated in **bold letters,** our replies are given in *italics,* and text modified or added to the manuscript is given in blue.

General comments:

**It would be nice if the authors discuss global comparison against observations such as the MPHP or MPHP2 datasets for an extended period of time (one year or more). If this was not done, then maybe it could be mentioned in the perspective section.**

*Unfortunately, this is not possible. We simulated the 10 day period in July 2010. On the MPHP/MPHP2 website there are only datasets available for 2001-2008/01.2008-10.2009. For a global comparison we would need global simulations. This could be done in future work after integrating the plume rise model into ICON-ART.*

**The impact on the horizontal diffusion of the plume (as compared to MODIS AOD observations, for example) has been less studied: the authors could maybe show a plot to describe this aspect.**

*We added additional text and a new figure to address that point.*

To evaluate the horizontal diffusion of the plume the simulated AOD is compared with AOD satellite retrievals, both at 550 nm. In the top of Fig. 8 observations made by MODIS on-board Terra and retrieved with the dark target algorithm are displayed time averaged over 14 and 15 July 2010. Below the AOD averaged over the four overpass times of Terra satellite are shown for the different simulations. The observed maximum of over 3.5 is located around 57.5° N, 112.5° W. From there the increased AOD is spread towards north-east and south-east. In all simulations the maximum is located slightly further in the east compared to the satellite retrieval. The pattern of AOD differs between all simulations in its width, shape, and strength. The southern extension of the plume reaching 50° N, 105° W is best represented by the simulations VARHEIGHT and 800M. Due to the coarse resolution of the satellite retrieval it is not possible to determine the overall best match.

[Figure]

**Figure 8.** AOD at 550 nm averaged over 14-15 July 2010. Top: Satellite retrieval from MODIS on-board Terra, below: Simulations VARHEIGHT, EMISSIONCYCLE, 800M, and 7500M.

**However, the other sources of uncertainties (turbulent diffusion, transport), etc... have not really been mentioned. While it is a hard job to estimate these sources of error, maybe a comparison of the forecasted meteorological parameters against observations (weather stations, reanalysis or radio-soundings if any radio-sounding is available in this area) could help.**

*The figures below allow a comparison of the simulated meteorological parameters and reanalysis from CFS (Climate Forecast System, available at [www.wetter3.de](www.wetter3.de)) for 2 m temperature and surface pressure. Therefore, simulation VARHEIGHT is used with a lead time of 36 hours. Simulation and reanalysis show reasonable agreement. For both the maximum temperature occurs in the northwestern part of the simulation domain with about 28 °C and a minimum temperature over the Hudson Bay with slightly over 0 °C. The ridge in the central southern part of the simulation domain is indicated by high pressure at the surface of more than 1010 hPa in the simulation and in the reanalysis.*

[Figure]

*Figure: Comparison of reanalysis from CFS (Climate Forecast System, available at [www.wetter3.de](www.wetter3.de)) and simulation VARHEIGHT for 11 July 2010 18 UTC and the meteorological variables 2 m temperature and surface pressure. (a) 2 m temperature [°C] denoted by the color coding and grey lines (CFS), (b) 2 m temperature [°C] denoted by the color coding (COSMO-ART), (c) surface pressure [hPa] displayed by the white lines (CFS), and (d) surface pressure [hPa] represented by the white lines and the color coding (COSMO-ART).*

**Maybe plot 13 could also be enlarge as well.**

*Done*

Specific comments:
• **Page 3 line 5 "Additional buoyancy can be gained through release of latent heat": for large fires latent heat can be an important contribution (pyro-CU and Cb)**

*Here we added:*
The release of latent heat from large fires can make an important contribution to the formation of pyrocumulus and pyrocumulonibus clouds (Fromm et al., 2010).

• **Page 3 lines 10-25: see the review of Paugam et al 2015: Paugam, R., Wooster, M., Freitas, S., and Val Martin, M.: A review of approaches to estimate wildfire plume injection height within large-scale atmospheric chemical transport models, Atmos. Chem. Phys., 16, 907-925, doi:10.5194/acp-16-907-2016, 2016.**

*We added:*
A recent review of the representation of plume injection heights in atmospheric models was performed by Paugam et al. (2016).

• **Page 5 line 19 "To demonstrate the importance of meteorological conditions on the maximum height of the plume top": indeed, sometimes the meteorological conditions can have more impact on the plume top height than the fire itself. In our experience with a later version of Freitas's PRM, the values for median injection height were sometimes higher with no fire forcing at the base than with fire forcing, which is anomalous (this happened in around 10% of cases with Aqua/Terra pixels). The authors are encouraged to test this kind of occurrence.**

*This is not the case in our version of the plume rise model. When the heat flux is set to zero we obtain zero plume height. We tested it for all plume conditions within our simulation domain and period.*

• **Page 6: Since Freitas's PRM provides a detrainment profile, I don't understand why the vertical distribution of emissions has to be parameterized in such a way. Instead of getting just the lower and upper bounds from the PRM, isn't it possible to get the whole detrainment profile and then interpolate it to COSMO-ART levels? Otherwise, the proposed parameterization seems sensible.**

*We do use the detrainment profile specified in Freitas' PRM. Since the levels of PRM are not infinitely small, it is more accurate to distribute the emissions within COSMO-ART over the height levels instead of interpolating emissions from discrete intervals from PRM to COSMO-ART levels. We just introduce the dimensionless height $z^* \geq 0$ and $\leq 1$ instead of the absolute height in this equation.*
*To clarify this we changed the sentence: The emissions are distributed with a parabolic function defined between the upper and the lower bounds as specified within the plume rise model and according to the following expression.*

• **Page 7, diurnal cycle section: the approach is alright. On this subject you can also refer to Andela, N., Kaiser, J. W., van der Werf, G. R., and Wooster, M. J.: New fire diurnal cycle characterizations to improve fire radiative energy assessments made from MODIS observations, Atmos. Chem. Phys., 15, 8831-8846, doi:10.5194/acp-15-8831-2015, 2015.**

*We included the citation.*

• **Page 8, Model configuration: There is a new GFAS dataset, GFASv1.2, which includes "mean heights of maximum injection" (the average of the PRM levels where detrainment is above half of the maximum detrainment) and "plume top", computed by the PRM from Freitas (and updated by R. Paugam) using MODIS observations and ECMWF meteorological profiles. It also includes**

**injection heights computed following Sofiev et al. (2012). It would be interesting for you to compare these data with the plume top that you obtained with the PRM.**

*We now show the comparison between the GFAS plume heights and the simulated ones (see comment below).*

• **Page 9 Plume heights: Since it is the main subject of the paper, it would be nice to have more information on the plume heights provided by COSMO-ART-PRM, maybe the top and bottom of the plume at some selected locations and times for example, or the emission profiles used in COSMO-ART.**

*We added a figure which shows the time series for one location having coincidental values for simulated plume height and values within GFAS plume heights datasets.*

[Figure]

Figure 5. Time series of the plume height at one fire location. The lower and the upper bound of the effective emission layer simulated by the plume rise model within COSMO-ART (blue lines). The dashed light blue line gives the daily mean of these heights. Plume heights specified by GFASv1.2 for two derivations (red and green line).

*In section 3.3 we added:*
In Fig. 5 the time series of the plume height is shown for one fire location (56.98° N, 106.99° W). The top blue line denotes the plume top and the lower blue line the plume bottom as simulated by COSMO-ART in combination with the plume rise model. Thereby the upper and the lower bound of the effective emission layer are defined. The diurnal cycle is clearly visible. The thickness of the smoke layer is dependent on the meteorological conditions. During night the smoke is located within 1 km above ground. Daytime values range from about 2 – 7 km for the simulation period. A daily mean is calculated over each day (LST) and averaged over plume top and bottom to obtain a quantity comparable to GFASv1.2 plume height derivations. These are namely the height of maximum injection derived by a later version of the plume rise model within C-IFS (Composition-Integrated Forecasting System) and the plume top estimated after a method by Sofiev et al. (2012). In comparison to the plume heights obtained by simulation VARHEIGHT the GFAS plume heights do not have a diurnal cycle. The daily mean plume height of simulation VARHEIGHT agrees with the GFAS heights in the same extend than they do to each other. According to the GFAS plume height derivations two short fire periods occurred during 10. - 19. July 2010 while the fire in simulation VARHEIGHT lasts for ten days.

**Literature:**

Andela, N., J. W. Kaiser, G. R. van der Werf, and M. J. Wooster, 2015: New fire diurnal cycle characterizations to improve fire radiative energy assessments made from modis observations. Atmos. Chem. Phys., 15 (15), 8831–8846, doi:10.5194/acp-15-8831-2015.

Fromm, M., D. T. Lindsey, R. Servranckx, G. Yue, T. Trickl, R. Sica, P. Doucet, and S. Godin-Beekmann, 2010: The untold story of pyrocumulonimbus. Bull. Am. Meteorol. Soc., 91 (9), 1193.

Paugam, R., M. Wooster, S. Freitas, and M. Val Martin, 2016: A review of approaches to estimate wildfire plume injection height within large-scale atmospheric chemical transport models. Atmos. Chem. Phys., 16 (2), 907–925, doi:10.5194/acp-16-907-2016.

---

## Author Comment (AC2) · 31 May 2016

We thank referee #3 for his valuable comments and suggestions. We followed them as explained below.

The reviewers comments are repeated in **bold letters,** our replies are given in *italics,* and text modified or added to the manuscript is given in blue.

GENERAL REMARKS

**The paper would be worthwhile for publication in ACP if the following major remarks were properly taken into account, in particular:**
**a sound argumentation on the choice of several parameters (in particular fire intensity),**
**a better description on how optical properties and aerosol-cloud interactions are calculated in the model.**

*Both points are captured in detail in the following.*

MAJOR REMARKS

**Section 2.1, page 5, lines 13 – 20. The parameters used in this study to obtain the lower and upper bounds of the plume height need to be much better justified, in the context of available fire studies. When reading the paper, one could think that the two upper and lower values given for fire intensity (30 and 80 kw/m2) represent a common range of observed values. But then these values are not really used in a statistical sense, but rather in a deterministic way to calculate lower and upper plume heights for a given fire. Isn't there a conceptual mismatch. Also the values chosen for the limiting vertical velocity and default fire size need justification. For all values, how would altering them with respect to their estimated uncertainty ranges alter the results of this paper. Some sensitivity tests would be welcome here.**

*The methodology for using a range of values for heat flux and sensitivity tests made with the plume rise model is described in Freitas et al. (2007). The application of a range of heat flux is justified not only by the variability associated with the vegetation condition, which is not known, but also by the own dynamic variation during the combustion process. Besides, this range is also applied in a statistical sense since the net emission in the 3-D atmospheric transport model might be associated not with a unique fire but a set of sub-grid scale fires all burning inside the same model grid box. Using the fire radiative power (FRP) to estimate the buoyancy flux does not help to eliminate the use of the prescribed range of the heat flux, since there is still a substantial uncertainty in converting FRP to the convective energy, which has been widely described in the literature (Wooster et al., 2005, Val Martin et al., 2012, Paugam et al., 2015). Moreover, the uncertainty in the FRP retrieval by sensors on-board of satellites is also high.*

**Section 2.4, page 7: This section is difficult to read, because the aim of the argumentation is not clear from the beginning on. The last sentence, that the authors were unable to perform Mie calculations for this study, and thus took values for diesel soot instead of wood soot should be put right in the beginning of the section. Potential implications of this approximation should be discussed all along the paper, in particular in section 3.5 (radiative effects).**

*We have completely rewritten this section following the reviewers' suggestions. Unfortunately, our original statement regarding the single scattering albedo of biomass burning aerosol and pure diesel soot particles was misleading. In COSMO-ART soot particles are subject to aging during the transport process and therefore also the single scattering albedo of the simulated aerosol population increases.* Calculating the radiative effect of biomass burning and other aerosol types requires the optical properties extinction coefficient, single scattering albedo, and asymmetry parameter of the aerosol particles at each grid point and each time step. These optical properties depend on the refractive

index of the individual compounds, the chemical composition of the particles, their shape, and their size distribution. The refractive index and therefore the optical properties depend on the wavelength.

Insoluble light absorbing particles like soot can be covered by a soluble shell due to physical (coagulation, condensation), and photo-chemical ageing. This increases their mass absorption efficiency (Riemer et al., 2003; Saleh et al., 2014; Bond et al., 2013). That effect needs to be accounted for in fully online-coupled model systems like COSMO-ART. Mie-calculations are the adequate method to determine the optical properties from given size distributions and their chemical composition (Bohren and Huffman, 2004). These calculations are very time consuming and therefore it is not possible to perform them at each grid point and at each time step. Instead, we have developed a parameterization as described in Vogel et al. (2009). This parameterization is based on simulated aerosol distributions and detailed Mie-calculations ending in mass specific values of the extinction coefficient, single scattering albedo and asymmetry parameter. Moreover, this parameterization takes into account the physical and chemical ageing of soot particles (Riemer et al., 2004; Vogel et al., 2009). Values are delivered for the wavelength bands of the radiation scheme used in COSMO-ART (Ritter and Geleyn, 1992). Fundamental input data for the Mie-calculations are the wavelength dependent refractive indices for the individual compounds. Here, we are using data of detailed measurements performed in the AIDA (Aerosol Interaction and Dynamics in the Atmosphere) chamber (Schnaiter et al., 2003). The disadvantage of this data is that it was obtained for pure diesel soot. But its advantage is the high spectral resolution of the data which is not the case for other lab studies. A comparison of this fundamental input data with data obtained for biomass burning aerosol is difficult for several reasons. Recent studies ended up with bulk data for mostly aged particles or with mass specific values for extinction and absorption coefficients. Consequently, quite different values were found depending on the specific burning conditions and particle compositions. In many cases values were gained for a single wavelength. For that reason it is hard to quantify the errors due to the calculation of the optical properties within COSMO-ART.

Following our parameterization we get a value for the mass extinction efficiency of 9.0 $m^2 g^{-1}$ for the spectral range 0.25 - 0.7 µm and for pure soot particles. For the soot containing Aitken mode we get a value of 5.0 $m^2 g^{-1}$, and for the soot containing accumulation mode a value of 4.0 $m^2 g^{-1}$. Laser measurements at a wavelength of 0.632 µm suggest a value of 7.8 $m^2 g^{-1}$ for soot with wood origin (Colbeck et al., 1997).

Levin et al. (2010) carried out measurements with biomass burning aerosol of different chemical composition. The geometric mean diameters ranged from 0.2 - 0.57 µm. For those particles they found refractive indices ranging from 1.55 - 1.80 for the real part and 0.01 - 0.50 for the imaginary part. They obtained dry mass extinction efficiency ranging from 1.64 - 6.64 $m^2 g^{-1}$ at a wavelength of 0.532 µm. Hungershoefer et al. (2008) found mass extinction efficiencies in the order of 9.0 $m^2 g^{-1}$ for savanna grass and African hardwood.

From these numbers we would conclude that the optical properties we are using are within the range of literature data.

**Fire aerosol is also constituted of organic aerosol. Which optical properties are adopted for organic aerosol? Is internal or external mixing assumed for different fire aerosol components? This should be stated. Only one reference for one wavelength is given for the single scattering albedo of diesel and wood. I guess that there are much more results available in literature. Please synthesize. Optical parameters of soot have been shown to change with plume age (for example review of Bond et al., 2013). This effect is not considered in the present study. This point should at least be discussed. Please also discuss, how specific information on size distribution would ideally be used for Mie calculations, and how this was handled in the present study. Again, what is the expected error?**

*This comment is addressed within the new version of section 2.4. Regarding the aging process we were not clear enough within the manuscript. In comparison to many other models it is a great advantage of COSMO-ART that it treats the aging of soot particles explicitly. Soot is treated as an*

*external mixture after its emissions and is then transferred by coagulation and chemical aging into an internal mixture (Riemer et al., 2003; Riemer et al., 2004).*

**Additional section 2.5: Please describe, how aerosol microphysics interactions are treated in the model, which processes and parameterizations are included? This is crucial for enabling the reader to understand results presented in Section 3.5 (Aerosol radiative impact).**

*The general model description was extended by specifications for the aerosol radiation interactions and aerosol cloud interactions.*

The simulations are conducted using the comprehensive online-coupled model system COSMO-ART (Consortium for Smallscale Modelling - Aerosols and Reactive Trace gases, Vogel et al., 2009). This system is based on the operational weather forecast model COSMO (Baldauf et al., 2011). COSMO-ART includes a comprehensive chemistry module to describe the gaseous composition of the atmosphere and secondary aerosol formation, and it allows for feedback of the simulated aerosol particles with radiation, cloud formation, and precipitation (Stanelle et al., 2010; Knote et al., 2011; Bangert et al., 2012; Lundgren et al., 2013; Athanasopoulou et al., 2014; Rieger et al., 2014; Vogel et al., 2014). The size distribution of aerosol within COSMO-ART is approximated by log-normal distributions. In Table 1, all required modes with their initial median diameters, standard deviations and chemical compositions are presented. The standard deviation is maintained constant while the median diameter of the aerosol changes during transport. Chemical reactions are calculated with RADMKA (Regional Acid Deposition Model Version Karlsruhe, Vogel et al., 2009) which is based on RADM2 (Regional Acid Deposition Model, Stockwell et al. 1990).The formation of secondary organic aerosol in calculated by a VBS approach (volatility basis set, Athanasopoulou et al. 2012). COSMO-ART explicitly treats the aging of soot particles transferring them from external to internal mixtures as described in Riemer et al. (2003). The radiative fluxes are calculated with the GRAALS radiation scheme (Ritter and Geleyn, 1992). Preliminary Mie-calculations have been performed for the initial aerosol particle size distributions and their chemical composition to obtain mass specific values for the extinction coefficient, single scattering albedo, and asymmetry parameter. These coefficients also depend on wavelength. To consider the optical properties of the current aerosol distribution the mass specific parameters obtained by the Mie-calculation are weighted with the mass fraction of the chemical components. Within COSMO-ART a full two-moment cloud microphysics scheme (Seifert and Beheng, 2006) is used. Aerosol activation is considered according to Fountoukis and Nenes (2005). Ice nucleation is based on the parameterization by Barahona and Nenes (2009a, b).

**Section 3.4: The arguments given for stating that the VARHEIGHT simulation is the best are to some extent convincing. Nevertheless, the given data set is quite restricted, are there more observations available? For instance in-situ PM measurements at surface sites? MODIS or POLDER AOD fields? Is it possible to put the discussion on a more quantitative basis (for example by calculation of correlation coefficients between simulations and observations?**

*We added a comparison with MODIS AOD fields (see comment below).*

**It should be mentioned while discussing results in section 3.4, that differences between simulations and observations could be due also to errors in fire intensity and emissions. In how far do such errors prohibit from drawing conclusions on the different plume rise schemes.**

*We added:*
Note that if errors are made in the estimate of fire intensity and emissions this will influence the concentration in all simulations, while the plume height is only affected in VARHEIGHT and EMISSCYCLE.

**Overall, section 3.4 is quite difficult to follow, may be it is possible to simplify, and not give all numbers. Those could be grouped together in a table.**

*We shortened this paragraph, and modified the corresponding figure.*

The most prominent features of the observed smoke distribution are marked with dark green circles. Circle A indicates smoke observed by CALIOP between 6 and 7.5 km altitude. This feature is well represented by the simulations 7500M and EMISSCYCLE, moderately represented in VARHEIGHT and 800M fails at this point. Circle B refers to smoke within the lowest 3.5 km. In all simulations the smoke is located a little lower at this position but each of them showing distinct patterns in each case. Circle C and the descending line represent the skewness of the smoke layer between 56 and 50° N. The decline seems to be stronger in the simulations than in the observations. The height is matched by simulation VARHEIGHT and 800M. In EMISSCYCLE the height is slightly overestimated and in 7500M the height is remarkably overestimated.

[Figure]

**Figure 9.** Cross section of aerosol subtypes of CALIPSO overpass at around 9:20 UTC 16 July 2010 (a). The black colour coding denotes the presence of smoke; brown, green, and red represents polluted dust, clean continental, and polluted continental, respectively. Cross section along the same CALIPSO track for simulations 800M (b), 7500M (c), VARHEIGHT (d) and EMISSCYCLE (e), here only soot concentrations greater than 0.01 $\mu g\ m^{-3}$ are displayed.

**Section 3.5 would be strengthened, if simulated effects on short wave radiation, temperature and cloud cover could be substantiated by observations, for the given case study. This should be possible from meteorological in situ and satellite observations.**
**Without observations, this section remains rather speculative.**

*We now included station measurements of the short wave radiation at Fort Smith:*

Observations of the global solar radiation at Fort Smith (60.01° N, 111.57° W, Meteomanz.com) do support these simulations. On 15 July 2010 at 6 UTC the station reports 1115 J cm$^{-2}$ during the last 24 hours. The simulation VARHEIGHT which includes the fire emissions yields 1029 J cm$^{-2}$ for the same 24 hour period, whilst the simulation NOFIRE results in 2222 J cm$^{-2}$. This is a typical value for cloudless, smoke-free days. For example on 11 July 2010 a value of 2168 J cm$^{-2}$ was reported at that station.

*In addition a satellite retrieval of AOD is added, this is specified in more detail later on.*

MINOR REMARKS

**Page 3, lines 7-9: Is this rapid transport to Europe due to prior vertical lifting into the upper troposphere with stronger winds. Please make this link clear in the revised text.**

*At this point we added:* This is due to lifting into high altitudes by pyro-convection prior to horizontal advection over the North Atlantic Ocean.

**Page 3, lines 19-23: are these arguments valid for specific cases or are they more general, please make this clear.**

*We only want to refer to their observations without any assumptions on generality.*
*We added:* in their case

**Page 4, lines 11-19: please better argue, why this study is new with respect to older work.**

To the best of our knowledge we are the first to investigate the effect of biomass burning aerosol on temperature and dynamics with an online-coupled modelling system on synoptic time scales with an explicit treatment of the aging of soot in combination with a plume rise model.

**Page 4, model description: Is secondary aerosol formation from biomass burning emissions included in the model? This process is for example shown to be important for Russian fires in summer 2010 (Konovalov et al., 2015).**

*We have extended section 2 to give a more clear and comprehensive description of COSMO-ART regarding the aerosol treatment. The VBS scheme included in COSMO-ART is described in Athanasopoulou et al. (2012).*

**Page 5, Section 2.1: is lateral detrainment in the convective fire plume is apparently not considered?**
*In our version of the plume rise model only entrainment of environmental air into the plume is considered. In a more recent version of the plume rise model detrainment was included (Paugam et al., 2015).*

**Page 7, line 18: 'in sufficient agreement' agreement with what?**

*This sentence has been removed from the text.*

**Page 11, line 32: A median mass diameter above 1µg/m3 seems large to me. It is for instance larger than the accumulation mode in which most mass of continental aged pollution aerosol is concentrated. Is there an explanation, why this is different for fire aerosol.**

*Since mass size distributions for biomass burning aerosol seem to be very rare, we decided to replace it by a comparison with a number size distribution from a laboratory measurement.*
The simulated number distributions for Fort Smith (60.01° N, 111.57° W; Fig. 3), a location in the fire (61.30° N, 110.45° W), and a location in the vicinity of the fire (58.12° N, 106.51° W) near the surface

on 15 July 2010 at 18:00 UTC are shown in Fig. 11. Unfortunately, we have no in-situ characterization of the aerosol particles. Instead we compare the model results with the size distribution measured during a small-scale laboratory experiment performed by Hungershoefer et al. (2008). For their experiment savanna grass and African hardwood were burnt in a smoke chamber in order to characterize the optical properties of biomass burning aerosol. At Fort Smith the simulated number concentration is about three orders of magnitude smaller than in the laboratory measurement, while the median diameter is about 0.1 µm in both cases. At the fire the simulated number concentration is comparable to the measurement but the simulated median diameter of 0.04 µm is smaller than in the measurement. Close to the fire the concentration gets smaller and the median diameter bigger than at the fire. The diameter is still smaller and the concentration still higher than in Fort Smith which is located further away from the fire. The aging process clearly arises out of the increasing median diameter with distance to the fire. Especially close to the fire the course of measurement and simulation show reasonable agreement. A smaller number concentration can be expected due to dispersion of fresh air outside a laboratory.

[Figure]

Figure 11. Number size distribution (a) at 60.01° N, 111.57° W (Fort Smith), (b) 61.30° N, 110.45° W (at fire), and (c) 58.12° N, 106.51° W (close to fire) at the surface on 15 July 2010 at 18:00 UTC for VARHEIGHT (red line) and as comparison measurements from an experimental fire (doted black line) performed by Hungershoefer et al. (2008).

**Tables:**
**It would be worthwhile to add a table with emission factors for different model species.**

*We added a new table which contains the emitted species and their assignment to the existent COSMO-ART classes. Furthermore, we added the following sentence.*
The species are listed in Table 2 together with their assignments and individual weightings, where necessary.

**Table 2.** Emitted gaseous and particulate species derived from GFASv1.1

| original notation | COSMO-ART class |
| --- | --- |
| Carbon Monoxide | CO |
| Nitrogen Oxides $NO_x * 0.9$ | NO |
| Nitrogen Oxides $NO_x * 0.1$ | NO2 |
| Sulfur Dioxide | SO2 |
| Ammonia ($NH_3$) | NH3 |
| Ethane ($C_2H_6$) | ETH (Ethan) |
| Methanol ($CH_3OH$) | HC3 ($C_3$ to $C_5$ Alkanes) |
| Ethanol ($C_2H_5OH$) | HC3 ($C_3$ to $C_5$ Alkanes) |
| Propane ($C_3H_8$) | HC3 ($C_3$ to $C_5$ Alkanes) |
| Butanes ($C_4H_{10}$) | HC3 ($C_3$ to $C_5$ Alkanes) |
| Pentanes ($C_5H_{12}$) | HC5 ($C_6$ to $C_8$ Alkanes) |
| Hexanes ($C_6H_{14}$) | HC5 ($C_6$ to $C_8$ Alkanes) |
| Heptane ($C_7H_{16}$) | HC8 (higher Alkanes) |
| Ethene ($C_2H_4$) | OL2 (Ethene) |
| Propene ($C_3H_6$) | OLT (terminal Alkenes) |
| Butenes ($C_4H_8$) | OLT (terminal Alkenes) |
| Octene ($C_8H_{16}$) | OLT (terminal Alkenes) |
| Pentenes ($C_5H_{10}$) | OLI (internal Alkenes) |
| Hexene ($C_6H_{12}$) | OLI (internal Alkenes) |
| Isoprene ($C_5H_8$) | ISO (Isoprene) |
| Terpenes ($C_5H_8$)n | API (Terpenes) |
| Toluene ($C_7H_8$) | TOL (Toluene) |
| Benzene ($C_6H_6$) | TOL (Toluene) |
| Xylene ($C_8H_{10}$) | XYL (Xylene) |
| Formaldehyde ($CH_2O$) | HCHO |
| Acetaldehyde ($C_2H_4O$) | ALD (Acetaldehyde) |
| Acetone ($C_3H_6O$) | KET (Ketones) |
| Black Carbon | s (pure soot mode) |
| Organic Carbon $* 0.1$ | if (Aitken mode particles, soot free) |
| Organic Carbon $* 0.9$ | jf (Accummulation mode particles, soot free) |

**Figures:**
**Figure 2: please specify the Figure legend, for which parameter the diurnal cycle is shown?**

*We changed the figure legend. It now reads as:*
The course of the diurnal cycle assumed for fires in boreal forests. In the individual simulations this diurnal cycle is overlaid on the daily values of fire size, fire intensity and emission strength.

**Figure 4: What is the meaning of the red points? I guess the smoke area is in grey, while clouds are white. To be completely clear, this could be mentioned in the legend.**

*We added:*
The red dots denote the fire locations. The grey structures state the distribution of smoke.

**How does observed smoke region compare to that simulated with different plume height options. Does such a comparison allow state on benefits of different plume height treatments?**

*In order to address this point we added an additional figure and the following text.*
To evaluate the horizontal diffusion of the plume the simulated AOD is compared with AOD satellite retrievals, both at 550 nm. In the top of Fig. 8 observations made by MODIS on-board Terra and retrieved with the dark target algorithm are displayed time averaged over 14 and 15 July 2010. Below the AOD averaged over the four overpass times of Terra satellite are shown for the different simulations. The observed maximum of over 3.5 is located around 57.5° N, 112.5° W. From there the increased AOD is spread towards north-east and south-east. In all simulations the maximum is located further in the east than in the satellite retrieval. The pattern of AOD differs between all simulations in its width, shape, and strength. The southern extension of the plume reaching 50° N, 105° W is best represented by the simulations VARHEIGHT and 800M. Due to the coarse resolution of the satellite retrieval it is not possible to determine the overall best match.

[Figure]

**Figure 8.** AOD at 550 nm averaged over 14-15 July 2010. Top: Satellite retrieval from MODIS on-board Terra, below: Simulations VARHEIGHT, EMISSIONCYCLE, 800M, and 7500M.

**Figure 5: It is difficult to make the "geographical" link between figures 4 and 5. In figure 5, please indicate latitudes and longitudes, or make appear the domain of fig. 4 in fig. 5.**

*The simulation domain shown in Fig. 3 and Fig. 4 is now indicated in Fig. 5. We added to the figure legend:*
The edges of the simulation domain are indicated with blue triangles.

**Figure 7: How are colors attributed, it is not very quantitative?**

*No, it is not quantitative. The aerosol subtypes are determined with an associated aerosol lidar ratio at 532 nm and 1064 nm. With this only a classification of the aerosol is treated.*

**This is mentioned in the main text, please recall it in the figure legend.**

*We added:*
The black colour coding denotes the presence of smoke; brown, green, and red represents polluted dust, clean continental, and polluted continental, respectively.

**Figure 10: Aren't there any observations of short wave radiation available in the modelling domain?**

*We now included station measurements of the short wave radiation at Fort Smith:*
Observations of the global solar radiation at Fort Smith (60.01° N, 111.57° W, Meteomanz.com) do support these simulations. On 15 July 2010 at 6 UTC the station reports 1115 J cm$^{-2}$ during the last 24 hours. The simulation VARHEIGHT which includes the fire emissions yields 1029 J cm$^{-2}$ for the same 24 hour period, whilst the simulation NOFIRE results in 2222 J cm$^{-2}$. In turn this value is a normal finding for cloudless, smoke-free days, e.g. on 11 July 2010 a value of 2168 J cm$^{-2}$ was reported at that station.

**Is figure 10 contained in Figure 13, or is it different. It could be justified, but please indicate it.**

*Yes, simulation VARHEIGHT is depicted in both figures. We put these two figures into one.*

TECHNICAL, EDITING REMARKS
**Page 2, line 7: 'they developped' -> 'Konovalov et al.' or 'the authors'**

*Done*

**Page 3, line 18: 'Another simulation....' In the same study/reference?**

*We changed another to* the. *This simulation was performed in the same study.*

**Page 3, line 18: 'the same' Which ?**

The source height of 800 m was meant. We added: of 800 m

**Page 6, line 18: 'the wind speed in the boundary layer is usually higher....' Add 'usually'**

*Done*

**REFERENCES:**

Athanasopoulou, E., H. Vogel, B. Vogel, A. Tsimpidi, S. N. Pandis, C. Knote, and C. Fountoukis, 2012: Modeling the meteorological and chemical effects of secondary organic aerosols during an EUCAARI campaign. Atmos. Chem. Phys., 13, 625.

Barahona, D. und A. Nenes, 2009a: Parameterizing the competition between homogeneous and heterogeneous freezing in cirrus cloud formation–monodisperse ice nuclei. Atmospheric Chemistry and Physics, 9 (2), 369–381.

Barahona, D. und A. Nenes, 2009b: Parameterizing the competition between homogeneous and heterogeneous freezing in ice cloud formation–polydisperse ice nuclei. Atmospheric Chemistry and Physics, 9 (16), 5933–5948.

Bohren, C. F. and D. R. Huffman, 2004: Absorption and scattering of light by small particles. Wiley, New York.

Bond, T. C., et al., 2013: Bounding the role of black carbon in the climate system: A scientific assessment. J. Geophys. Res.-Atmos., 118 (11), 5380–5552.

Fountoukis, C. und A. Nenes, 2005: Continued development of a cloud droplet formation parameterization for global climate models. Journal of Geophysical Research: Atmospheres (1984–2012), 110 (D11).

Hungershoefer, K., et al., 2008: Modelling the optical properties of fresh biomass burning aerosol produced in a smoke chamber: results from the EFEU campaign. Atmos. Chem. Phys., 8 (13), 3427–3439.

Levin, E., et al., 2010: Biomass burning smoke aerosol properties measured during Fire Laboratory at Missoula Experiments (FLAME). J. Geophys. Res.-Atmos., 115 (D18).

Paugam, R., Wooster, M., Atherton, J., Freitas, S. R., Schultz, M. G., & Kaiser, J. W. (2015). Development and optimization of a wildfire plume rise model based on remote sensing data inputs–Part 2. *Atmos. Chem. Phys. Discuss*, *15*, 9815-9895.

Riemer, N., H. Vogel, and B. Vogel, 2004: Soot aging time scales in polluted regions during day and night. Atmos. Chem. Phys., 4 (7), 1885–1893.

Ritter, B. and J.-F. Geleyn, 1992: A comprehensive radiation scheme for numerical weather prediction models with potential applications in climate simulations. Mon. Weather Rev., 120 (2), 303–325.

Seifert, A. und K. Beheng, 2006: A two-moment cloud microphysics parameterization for mixedphase clouds. Part 1: Model description. Meteorology and atmospheric physics, 92 (1-2), 45–66.

Stockwell, W. R., Middleton, P., and Chang, J. S.: The second generation regional acid deposition model chemical mechanism for regional air quality modelling, J. Geophys. Res., 95, 16343–16367, 1990.

Saleh, R., et al., 2014: Brownness of organics in aerosols from biomass burning linked to their black carbon content. Nat. Geosci., 7 (9), 647–650.

Schnaiter, M., H. Horvath, 5 O. Möhler, K.-H. Naumann, H. Saathoff, and O. Schöck, 2003: UV-VIS-NIR spectral optical properties of soot and soot-containing aerosols. J. Atmos. Sci., 34 (10), 1421–1444.

Val Martin, M., Kahn, R. A., Logan, J. A., Paugam, R., Wooster, M., & Ichoku, C. (2012). Space-based observational constraints for 1-D fire smoke plume-rise models. *Journal of Geophysical Research: Atmospheres*, *117*(D22).

Wooster, M. J., G. Roberts, G. L. W. Perry, and Y. J. Kaufman, 2005: Retrieval of biomass combustion rates and totals from fire radiative power observations: FRP derivation and calibration relationships between biomass consumption and fire radiative energy release. J. Geophys. Res.-Atmos., 110 (D24), doi:10.1029/2005JD006318, d24311.

---

## Author Comment (AC3) · 31 May 2016

We thank referee #1 for his valuable comments and suggestions. We followed them as explained below.

The reviewers comments are repeated in **bold letters,** our replies are given in *italics,* and text modified or added to the manuscript is given in blue.

Specific Comments:

**- Line 36: Please replace 'they' with 'Konovalov et al., 2015' to clarify the reference.**

*Done*

**- Line 136: Please add a reference for the used data set for the fire emissions; I guess Kaiser et al., 2012, would be appropriate when referring to GFAS, but maybe there is a more recent reference available.**

*We inserted the citation Kaiser et al.(2012).*

**- Section 2.3: The time step of the plume model calculations is not clearly stated in this section; I assume the emissions are updated with an hourly time step, but it would be useful to have this stated here (or in Section 3.1 in case the time step is flexible and can be easily adjusted according to the model simulation, similar to the integration time step).**

*The frequency of plume height calculations is given in 2.1: "Hourly, for every grid point with an active fire, the values of these variables are transferred to the plume rise model. Within an hour, the input variables are maintained constant."*
*To clarify this point we modified the text:*
Hourly, for every grid point with an active fire, the values of these variables are transferred to the plume rise model in order to calculate the current plume height. Within an hour, the plume height is maintained constant.
*And added in section 2.3:* Within an hour the emissions are maintained constant.

**- Section 2.4: The chemical composition and the optical properties of biomass burning aerosol are rather complex; to the best of my knowledge it is still uncertain whether fuel load and/or fire type (i.e, smoldering or flaming) determine the optical properties of the biomass burning aerosol. Hence, certain assumptions on the optical properties of the emitted aerosol have to be made in model simulations such as those presented here. However, please refer to and discuss some references dealing with the study of biomass burning aerosol and its optical properties, e.g., Hungershöfer et al., 2008; Levin et al., 2010; Saleh et al., 2014.**

*We have rewritten this section and added the proposed references:*
Calculating the radiative effect of biomass burning and other aerosol types requires the optical properties extinction coefficient, single scattering albedo, and asymmetry parameter of the aerosol particles at each grid point and each time step. These optical properties depend on the refractive index of the individual compounds, the chemical composition of the particles, their shape, and their size distribution. The refractive index and therefore the optical properties depend on the wavelength.
Insoluble light absorbing particles like soot can be covered by a soluble shell due to physical (coagulation, condensation), and photo-chemical ageing. This increases their mass absorption efficiency (Riemer et al., 2003; Saleh et al., 2014; Bond et al., 2013). That effect needs to be accounted for in fully online-coupled model systems like COSMO-ART. Mie-calculations are the adequate method to determine the optical properties from given size distributions and their

chemical composition (Bohren and Huffman, 2004).These calculations are very time consuming and therefore it is not possible to perform them at each grid point and at each time step. Instead, we have developed a parameterization as described in Vogel et al. (2009). This parameterization is based on simulated aerosol distributions and detailed Mie-calculations ending in mass specific values of the extinction coefficient, single scattering albedo and asymmetry parameter. Moreover, this parameterization takes into account the physical and chemical ageing of soot particles (Riemer et al., 2004; Vogel et al., 2009).Values are delivered for the wavelength bands of the radiation scheme used in COSMO-ART (Ritter and Geleyn, 1992). Fundamental input data for the Mie-calculations are the wavelength dependent refractive indices for the individual compounds. Here, we are using data of detailed measurements performed in the AIDA (Aerosol Interaction and Dynamics in the Atmosphere) chamber (Schnaiter et al., 2003). The disadvantage of this data is that it was obtained for pure diesel soot. But its advantage is the high spectral resolution of the data which is not the case for other lab studies. A comparison of this fundamental input data with data obtained for biomass burning aerosol is difficult for several reasons. Recent studies ended up with bulk data for mostly aged particles or with mass specific values for extinction and absorption coefficients. Consequently, quite different values were found depending on the specific burning conditions and particle compositions. In many cases values were gained for a single wavelength. For that reason it is hard to quantify the errors due to the calculation of the optical properties within COSMO-ART.

Following our parameterization we get a value for the mass extinction efficiency of $9.0\ m^2g^{-1}$ for the spectral range 0.25 - 0.7 µm and for pure soot particles. For the soot containing Aitken mode we get a value of $5.0\ m^2g^{-1}$, and for the soot containing accumulation mode a value of $4.0\ m^2g^{-1}$. Laser measurements at a wavelength of 0.632 µm suggest a value of $7.8\ m^2g^{-1}$ for soot with wood origin (Colbeck et al., 1997).

Levin et al. (2010) carried out measurements with biomass burning aerosol of different chemical composition. The geometric mean diameters ranged from 0.2 to 0.57 µm. For those particles they found refractive indices ranging from 1.55 to 1.80 for the real part and 0.01 - 0.50 for the imaginary part. They obtained dry mass extinction efficiency ranging from 1.64 to 6.64 $m^2g^{-1}$ at a wavelength of 0.532 µm. Hungershoefer et al. (2008) found mass extinction efficiencies in the order of 9.0 $m^2g^{-1}$ for savanna grass and African hardwood.

From these numbers we would conclude that the optical properties we are using are within the range of literature data.

**- Line 257 ff: Clearly the use of the single-scattering albedo for diesel soot results in an overestimation of the absorption of the emitted wildland fire aerosol, as correctly stated in the manuscript. Since the improved treatment of the optical properties of the emitted aerosol is not the main purpose of this study, it seems appropriate for the present work to use the aerosol classes available in the modeling system. However, for follow-up studies, in particular studies related to the dynamical feedback of the biomass burning aerosol on the atmosphere through aerosol absorption, this significant limitation of the model systems requires improvement. For the current study, please remove 'may slightly' from the final sentence of this paragraph so that it reads: 'Using the optical properties of diesel soot for our simulations, we overestimate the absorption in layers of dense smoke.'**

*We regret that our formulation concerning the single scattering albedo gave the impression that we are strongly overestimating the effect of biomass burning aerosol. For that reason we have rewritten section 2.4.*

**- Line 272: Please check whether the reference to Kaiser et al.., 2009a, can be replaced by referring to Kaiser et al., 2012, which is a peer-reviewed publication and not a Technical Document.**

*At this point we do not want to replace the reference since the Technical Document better describes the diurnal cycle. But in section 3.1 it is appropriate to replace Kaiser et al. (2009a) by Kaiser et al. (2012). This was done.*

**- Line 275: Please add some more information on the properties of the emitted aerosol particles; e.g., to which modes and composition the emitted aerosol particles are allocated. These classes could maybe be highlighted in Table 1.**

*We added a table which contains the emitted species and their assignment to the existent COSMO-ART classes.*
The species are listed in Table 2 together with their assignments and individual weightings, where necessary.

**Table 2.** Emitted gaseous and particulate species derived from GFASv1.1

| original notation | COSMO-ART class |
| --- | --- |
| Carbon Monoxide | CO |
| Nitrogen Oxides $NO_x * 0.9$ | NO |
| Nitrogen Oxides $NO_x * 0.1$ | NO2 |
| Sulfur Dioxide | SO2 |
| Ammonia ($NH_3$) | NH3 |
| Ethane ($C_2H_6$) | ETH (Ethan) |
| Methanol ($CH_3OH$) | HC3 ($C_3$ to $C_5$ Alkanes) |
| Ethanol ($C_2H_5OH$) | HC3 ($C_3$ to $C_5$ Alkanes) |
| Propane ($C_3H_8$) | HC3 ($C_3$ to $C_5$ Alkanes) |
| Butanes ($C_4H_{10}$) | HC3 ($C_3$ to $C_5$ Alkanes) |
| Pentanes ($C_5H_{12}$) | HC5 ($C_6$ to $C_8$ Alkanes) |
| Hexanes ($C_6H_{14}$) | HC5 ($C_6$ to $C_8$ Alkanes) |
| Heptane ($C_7H_{16}$) | HC8 (higher Alkanes) |
| Ethene ($C_2H_4$) | OL2 (Ethene) |
| Propene ($C_3H_6$) | OLT (terminal Alkenes) |
| Butenes ($C_4H_8$) | OLT (terminal Alkenes) |
| Octene ($C_8H_{16}$) | OLT (terminal Alkenes) |
| Pentenes ($C_5H_{10}$) | OLI (internal Alkenes) |
| Hexene ($C_6H_{12}$) | OLI (internal Alkenes) |
| Isoprene ($C_5H_8$) | ISO (Isoprene) |
| Terpenes ($C_5H_8$)n | API (Terpenes) |
| Toluene ($C_7H_8$) | TOL (Toluene) |
| Benzene ($C_6H_6$) | TOL (Toluene) |
| Xylene ($C_8H_{10}$) | XYL (Xylene) |
| Formaldehyde ($CH_2O$) | HCHO |
| Acetaldehyde ($C_2H_4O$) | ALD (Acetaldehyde) |
| Acetone ($C_3H_6O$) | KET (Ketones) |
| Black Carbon | s (pure soot mode) |
| Organic Carbon $* 0.1$ | if (Aitken mode particles, soot free) |
| Organic Carbon $* 0.9$ | jf (Accummulation mode particles, soot free) |

**- Line 311: What is the frequency of the plume height calculation used to generate Figure 6? Does the plume height represent the hourly emission height (i.e., every fire plume being counted multiple times) or the mean for each fire over a certain period (i.e., every fire counted only once). Please specify.**

*We clarified this:*
Thereby every plume top height calculated by the plume rise model is counted. If the fire is still active the plume is counted again in the next hour with its new height.

**- Line 324: Please replace 'through' by 'trough'**

*Done*

**- Line 353: Please start a new paragraph after '. . .aerosol type.'.**

*Done*

**- Line 354 – 382: This paragraph is rather hard to follow; from my perspective it contains too many numbers. The authors might consider to add a table with the corresponding numbers and to substantially shorten this section.**

*We shortened this paragraph, and inserted markers into the figure.*
The most prominent features of the observed smoke distribution are marked with dark green circles. Circle A indicates smoke observed by CALIOP between 6 and 7.5 km altitude. This feature is well represented by the simulations 7500M and EMISSCYCLE, moderately represented in VARHEIGHT and 800M fails at this point. Circle B refers to smoke within the lowest 3.5 km. In all simulations the smoke is located a little lower at this position but each of them showing distinct patterns in each case. Circle C and the descending line represent the skewness of the smoke layer between 56 and 50° N. The decline seems to be stronger in the simulations than in the observations. The height is matched by simulation VARHEIGHT and 800M. In EMISSCYCLE the height is slightly overestimated and in 7500M the height is remarkably overestimated.

[Figure]

**Figure 9.** Cross section of aerosol subtypes of CALIPSO overpass at around 9:20 UTC 16 July 2010 (a). The black colour coding denotes the presence of smoke; brown, green, and red represents polluted dust, clean continental, and polluted continental, respectively. Cross section along the same CALIPSO track for simulations 800M (b), 7500M (c), VARHEIGHT (d) and EMISSCYCLE (e), here only soot concentrations greater than 0.01 $\mu g\ m^{-3}$ are displayed.

**- Line 390 ff: The comparison with the data from the AERONET station at Bratts Lake is only performed for a single day (15 July). Would it be possible to repeat this analysis for other days, in particular for 16 July when the CALIPSO data are available. Please extend the comparison with available AERONET data from other days in July 2010.**

*The smoke passes the AERONET station at Bratts Lake only on 15 July. None of the few other stations in the simulation domain observe any smoke of this event.*

**- Section 3.5: Please clearly state at the beginning of this section the limitation of the analysis of the radiative impact of the biomass burning aerosol due to the use of the optical properties from diesel soot instead of biomass burning aerosol.**

*We added:*
*Uncertainties in the radiative impact of biomass burning aerosol are determined by the uncertainties in the description of its optical properties.*

**- Line 430 ff: Please motivate the use of Fort Smith to assess the aerosol impact on surface solar radiation. Obviously it would be very valuable if surface measurements would be available to**

**complement the comparison between the different model simulations. Are there corresponding measurements available at the AERONET site in Bratts Lake?**

*We now included measurements of the global solar radiation at the station Fort Smith:*
Observations of the global solar radiation at Fort Smith (60.01° N, 111.57° W, Meteomanz.com) do support these simulations. On 15 July 2010 at 6 UTC the station reports 1115 J cm$^{-2}$ during the last 24 hours. The simulation VARHEIGHT which includes the fire emissions yields 1029 J cm$^{-2}$ for the same 24 hour period, whilst the simulation NOFIRE results in 2222 J cm$^{-2}$. This is a typical value for cloudless, smoke-free days. For example on 11 July 2010 a value of 2168 J cm$^{-2}$ was reported at that station.
*Unfortunately such measurements are not available for Bratts Lake.*

**- Figure 11: It is striking that no temperature change is simulated around 106_W/ 58_N, despite the high aerosol loading as shown in Figure 4. Please comment.**

*We added this explanation:*
The lack of a cooling region is due to advection of heated air by cloud dissipation upstream the fires.

**- Line 475 / Figure 13: Move this paragraph and the figure towards Fig. 10 and the corresponding text.**

*Done*

**References:**

Bohren, C. F. and D. R. Huffman, 2004: Absorption and scattering of light by small particles. Wiley, New York.

Bond, T. C., et al., 2013: Bounding the role of black carbon in the climate system: A scientific assessment. J. Geophys. Res.-Atmos., 118 (11), 5380–5552.

Hungershoefer, K., et al., 2008: Modelling the optical properties of fresh biomass burning aerosol produced in a smoke chamber: results from the EFEU campaign. Atmos. Chem. Phys., 8 (13), 3427–3439.

Levin, E., et al., 2010: Biomass burning smoke aerosol properties measured during Fire Laboratory at Missoula Experiments (FLAME). J. Geophys. Res.-Atmos., 115 (D18).

Riemer, N., H. Vogel, and B. Vogel, 2004: Soot aging time scales in polluted regions during day and night. Atmos. Chem. Phys., 4 (7), 1885–1893.

Ritter, B. and J.-F. Geleyn, 1992: A comprehensive radiation scheme for numerical weather prediction models with potential applications in climate simulations. Mon. Weather Rev., 120 (2), 303–325.

Saleh, R., et al., 2014: Brownness of organics in aerosols from biomass burning linked to their black carbon content. Nat. Geosci., 7 (9), 647–650.

Schnaiter, M., H. Horvath, 5 O. Möhler, K.-H. Naumann, H. Saathoff, and O. Schöck, 2003: UV-VIS-NIR spectral optical properties of soot and soot-containing aerosols. J. Atmos. Sci., 34 (10), 1421–1444.